# REINFORCED IN-CONTEXT BLACK-BOX OPTIMIZATION

## ABSTRACT

Black-Box Optimization (BBO) has found successful applications in many fields of science and engineering. Recently, there has been a growing interest in meta-learning particular components of BBO algorithms to speed up optimization and get rid of tedious hand-crafted heuristics. As an extension, learning the entire algorithm from data requires the least labor from experts and can provide the most flexibility. In this paper, we propose RIBBO, a method to reinforce-learn a BBO algorithm from offline data in an end-to-end fashion. RIBBO employs expressive sequence models to learn the optimization histories produced by multiple behavior algorithms and tasks, leveraging the in-context learning ability of large models to extract task information and make decisions accordingly. Central to our method is to augment the optimization histories with *regret-to-go* tokens, which are designed to represent the performance of an algorithm based on cumulative regret over the future part of the histories. The integration of regret-to-go tokens enables RIBBO to automatically generate sequences of query points that are positive correlation to the user-desired regret, which is verified by its universally good empirical performance on diverse problems, including BBO benchmark functions, hyper-parameter optimization and robot control problems.

## 1 INTRODUCTION

Black-Box Optimization (BBO) Alarie et al. (2021); Audet & Hare (2017) refers to optimizing objective functions where neither analytic expressions nor derivatives of the objective are available. To solve BBO problems, we can only access the results of objective evaluation, which usually also incurs a high computational cost. Many fundamental problems in science and engineering involve optimization of expensive BBO functions, such as drug discovery Negoescu et al. (2011); Terayama et al. (2021), material design Frazier & Wang (2016); Gómez-Bombarelli et al. (2018), robot control Calandra et al. (2016); Chatzilygeroudis et al. (2019), and optimal experimental design Greenhill et al. (2020); Nguyen et al. (2023), just to name a few.

To date, a lot of BBO algorithms have been developed, among which the most prominent ones are Bayesian Optimization (BO) Frazier (2018); Shahriari et al. (2016) and Evolutionary Algorithms (EA) Back (1996); Zhou et al. (2019). Despite the advancements, these algorithms typically solve BBO problems from scratch and rely on expert-derived heuristics. Consequently, they are often hindered by slow convergence rates, and unable to leverage the inherent structures within the optimization problems Astudillo & Frazier (2021); Bai et al. (2023).

Recently, there has been a growing interest in meta-learning a particular component of the algorithms with previously collected data Arango et al. (2021); Feurer et al. (2021). Learning the component not only alleviates the need for the laborious design process of the domain experts, but also specifies the component with domain data to facilitate subsequent optimization. For example, some components in BO are proposed to be learned from data, including the surrogate model Müller et al. (2023); Perrone et al. (2018); Wang et al. (2021); Wistuba & Grabocka (2021), acquisition function Hsieh et al. (2021); Volpp et al. (2020), initialization strategy Feurer et al. (2015); Poloczek et al. (2016), and search space Li et al. (2022); Perrone & Shen (2019); some core evolutionary operations in EA have also been considered, e.g., learning the selection and mutation rate adaptation in genetic algorithm Lange et al. (2023a) or the update rules for evolution strategy Lange et al.

(2023b); the configuration of the algorithm can also be learned and dynamically adjusted throughout the optimization process Biedenkapp et al. (2020); Adriaensen et al. (2022).

There have also been some attempts to learn an entire algorithm in an End-to-End (E2E) fashion, which requires almost no expert knowledge at all and provides the most flexibility across a broad range of BBO problems. However, existing practices require additional knowledge regarding the objective function during the training stage, e.g., the gradient information (often impractical for BBO) Chen et al. (2017); TV et al. (2019) or online sampling from the objective function (often very expensive) Maraval et al. (2023). Chen et al. (2022) proposed the OptFormer method to imitate the behavior algorithms separately during training, presenting a challenge for the user to manually specify which algorithm to execute during testing. Thus, these methods are less ideal for practical scenarios where offline datasets are often available beforehand and a suitable algorithm for the given task has to be identified automatically without the involvement of domain experts.

In this paper, we introduce Reinforced In-context BBO (RIBBO), which learns a reinforced BBO algorithm from offline datasets in an E2E fashion. RIBBO employs an expressive sequence model, i.e., causal transformer, to fit the optimization histories in the offline datasets generated by executing diverse behavior algorithms on multiple tasks. The sequence model is fed with previous query points and their function values, and trained to predict the distribution over the next query point. During testing, the sequence model itself serves as a BBO algorithm by generating the next query points auto-regressively. Apart from this, RIBBO augments the optimization histories with *regret-to-go* (RTG) tokens, which are calculated by summing up the regrets over the future part of the histories, representing the future performance of an algorithm. A novel Hindsight Regret Relabelling (HRR) strategy is proposed to update the RTG tokens during testing. By integrating the RTG tokens into the modeling, RIBBO can automatically identify different algorithms, and generate sequences of query points that are positive correlation to the user-desired regret. Such modeling enables RIBBO to circumvent the impact of inferior data and further reinforce its performance on top of the behavior algorithms.

We perform experiments on BBOB synthetic functions, hyper-parameter optimization and robot control problems by using some representatives of heuristic search, EA, and BO as behavior algorithms to generate the offline datasets. The results show that RIBBO can automatically generate sequences of query points related to the user-desired regret across diverse problems, and achieve good performance universally. Note that the best behavior algorithm depends on the problem at hand, and RIBBO can perform even better on some problems. Compared to the most related method OptFormer Chen et al. (2022), RIBBO also has clear advantage. In addition, we perform a series of experiments to analyze the influence of important components of RIBBO.

## 2 BACKGROUND

### 2.1 BLACK-BOX OPTIMIZATION

Let $f : \mathcal{X} \to \mathbb{R}$ be a black-box function, where $\mathcal{X} \subseteq \mathbb{R}^d$ is a $d$-dimensional search space. The goal of BBO is to find an optimal solution $\boldsymbol{x}^* \in \arg\max_{\boldsymbol{x} \in \mathcal{X}} f(\boldsymbol{x})$, with the only permission of querying the objective function value. Several classes of BBO algorithms have been proposed, e.g., BO Frazier (2018); Shahriari et al. (2016) and EA Back (1996); Zhou et al. (2019). The basic framework of BO contains two critical components: a surrogate model, typically formalized as Gaussian Process (GP) Rasmussen & Williams (2006), and an acquisition function Wilson et al. (2018), which are used to model $f$ and decide the next query point, respectively. EA is a class of heuristic optimization algorithms inspired by natural evolution. It maintains a population of solutions and iterates through mutation, crossover, and selection operations to find better solutions.

To evaluate the performance of BBO algorithms, regrets are often used. The instantaneous regret $r_t = f(\boldsymbol{x}^*) - f(\boldsymbol{x}_t)$ measures the gap of function values between an optimal solution $\boldsymbol{x}^*$ and the currently selected point $\boldsymbol{x}_t$. The cumulative regret $\mathrm{Reg}_T = \sum_{i=1}^{T} r_i$ is the sum of instantaneous regrets in the first $T$ iterations.

## 2.2 META-LEARNING IN BLACK-BOX OPTIMIZATION

Hand-crafted BBO algorithms usually require an expert to analyze the algorithms' behavior across a wide range of problems, a process that is both tedious and time-consuming. One solution is meta-learning Vilalta & Drissi (2002); Hospedales et al. (2021), which aims to exploit knowledge to improve the performance of learning algorithms given data from a collection of tasks. By parameterizing a component of BBO algorithms or even an entire BBO algorithm that is traditionally manually designed, we can utilize historical data to incorporate the domain knowledge into the optimization, which may bring speedup.

**Meta-learning particular components** has been studied with different BBO algorithms. Meta-learning in BO can be divided into four main categories according to "what to transfer" Bai et al. (2023), including the design of the surrogate model, acquisition function, initialization strategy, and search space. For surrogate model design, Wang et al. (2021) and Wistuba & Grabocka (2021) parameterized the mean or kernel function of the GP model with Multi-Layer Perceptron (MLP), while Perrone et al. (2018) and Müller et al. (2023) substituted GP with Bayesian linear regression or neural process Garnelo et al. (2018); Müller et al. (2022). For acquisition function design, MetaBO Volpp et al. (2020) uses Reinforcement Learning (RL) to meta-train an acquisition function on a set of related tasks, and FSAF Hsieh et al. (2021) employs a Bayesian variant of deep Q-network as a surrogate differentiable acquisition function trained by model-agnostic meta-learning Finn et al. (2017). The remaining two categories focus on exploiting the previous good solutions to warm start the optimization Feurer et al. (2015); Poloczek et al. (2016) or shrink the search space Perrone & Shen (2019); Li et al. (2022). Meta-learning in EA usually focuses on learning specific evolutionary operations. For example, Lang et al. substituted core genetic operators, i.e., selection and mutation rate adaptation, with dot-product attention modules Lange et al. (2023a), and meta-learned a self-attention-based architecture to discover effective and order-invariant update rules Lange et al. (2023b). ALDes Zhao et al. (2024) introduces an auto-regressive learning-based approach to sequentially generate components of meta-heuristic algorithms. Beyond that, Dynamic Algorithm Configuration (DAC) Biedenkapp et al. (2020); Adriaensen et al. (2022) concentrates on learning the configurations of algorithms, employing RL to dynamically adjust the configurations during the optimization process.

**Meta-learning entire algorithms** has also been explored to obtain more flexible models. Early works Chen et al. (2017); TV et al. (2019) use Recurrent Neural Network (RNN) to meta-learn a BBO algorithm by optimizing the summed objective functions of some iterations. RNN uses its memory state to store information about history and outputs the next query point. This work assumes access to gradient information during the training phase, which is, however, usually impractical in BBO problems. OptFormer Chen et al. (2022) uses a text-based transformer framework to learn an algorithm, providing a universal E2E interface for BBO problems. It is trained to imitate different BBO algorithms across a broad range of problems, which, however, presents a challenge for the user to manually specify an algorithm for inference. Neural Acquisition Processes (NAP) Maraval et al. (2023) uses transformer to meta-learn the surrogate model and acquisition function of BO jointly. Due to the lack of labeled acquisition data, NAP uses an online RL algorithm with a supervised auxiliary loss for training, which requires online sampling from the expensive objective function and lacks efficiency. Black-box Optimization NETworks (BONET) Krishnamoorthy et al. (2023) employ a transformer model to fit regret-augmented trajectories in an offline BBO scenario, where the training and testing data are from the same objective function, and a prefix sequence is required to warm up the optimization before testing. OPT-GAN Lu et al. (2023) utilizes generative adversarial networks (GAN) to estimate the distribution of optimum gradually by exploration-exploitation trade-off. Compared to the above state-of-the-art E2E methods, we consider the meta-BBO setting, where the training datasets consist of diverse algorithms across different functions. Our approach offers the advantage of automatically identifying with RTG tokens and deploying the best-performing algorithm without requiring the user to pre-specify which algorithm to use or to provide a prefix sequence during the testing phase. It utilizes a supervised learning loss for training on a fixed offline dataset without the need for further interaction with the objective function.

## 2.3 DECISION TRANSFORMER

Transformer has emerged as a powerful architecture for sequence modeling tasks Khan et al. (2022); Wen et al. (2022); Wolf et al. (2020). A basic building block behind transformer is the self-attention

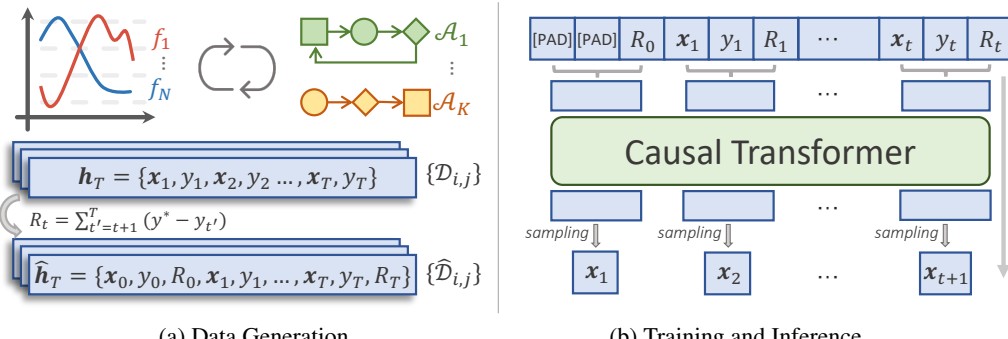

(a) Data Generation           (b) Training and Inference

Figure 1: Illustration of RIBBO. *Left: Data Generation.* $K$ existing BBO algorithms $\{\mathcal{A}_j\}_{j=1}^K$ and $N$ BBO tasks $\{f_i\}_{i=1}^N$ are used to serve as the behavior algorithms and the training tasks, respectively. The offline datasets $\{\mathcal{D}_{i,j}\}$ consist of the optimization histories $\boldsymbol{h}_T = \{(\boldsymbol{x}_t, y_t)\}_{t=1}^T$ collected by executing each behavior algorithm $\mathcal{A}_j$ on each task $f_i$ for $T$ evaluation steps, which are then augmented with the regret-to-go tokens $R_t$ (calculated as the cumulative regret $\sum_{t'=t+1}^T (y^* - y_{t'})$ over the future optimization history) to generate the final dataset $\{\widehat{\mathcal{D}}_{i,j}\}$ for training. *Right: Training and Inference.* Our model takes in triplets of $(\boldsymbol{x}_t, y_t, R_t)$, embeds them into one token, and outputs the distribution over the next query point $\boldsymbol{x}_{t+1}$. During training, the ground-truth next query point is used to minimize the loss in Eq. (4). During inference, the next query point $\boldsymbol{x}_{t+1}$ is generated auto-regressively based on the current history $\hat{\boldsymbol{h}}_t$.

mechanism Vaswani et al. (2017), which captures the correlation between tokens of any pair of timesteps. As the scale of data and model increases, transformer has demonstrated the *in-context learning* ability Brown et al. (2020), which refers to the capability of the model to infer the tasks at hand based on the input contexts. Decision Transformer (DT) Chen et al. (2021) abstracts RL as a sequence modeling problem, and introduces return-to-go tokens, representing the cumulative rewards over future interactions. Conditioning on return-to-go tokens enables DT to correlate the trajectories with their corresponding returns and generate future actions to achieve a user-specified return. Inspired by DT, we will treat BBO tasks as a sequence modeling problem naturally, use a causal transformer for modeling, and train it by conditioning on future regrets. Such design is expected to enable the learned model to distinguish algorithms with different performance and achieve good performance with a user-specified low regret.

## 3 METHOD

This section presents Reinforced In-context Black-Box Optimization (RIBBO), which learns an enhanced BBO algorithm in an E2E fashion, as illustrated in Figure 1. We follow the task-distribution assumption, which is commonly adopted in meta-learning settings Finn et al. (2017); Hospedales et al. (2021); Zhou et al. (2023). Our goal is to learn a generalizable model $\mathcal{M}$ capable of solving a wide range of BBO tasks, each associated with a BBO objective function $f$ sampled from the task distribution $P(\mathcal{F})$, where $\mathcal{F}$ denotes the function space.

Let $[N]$ denote the integer set $\{1, 2, \ldots, N\}$. During training, we usually access $N$ source tasks and each task corresponds to an objective function $f_i \sim P(\mathcal{F})$, where $i \in [N]$. Hereafter, we use $f_i$ to denote the task $i$ if the context is clear. We assume that the information is available via offline datasets $\mathcal{D}_{i,j}$, which are produced by executing a behavior algorithm $\mathcal{A}_j$ on task $f_i$, where $j \in [K]$ and $i \in [N]$. Each dataset $\mathcal{D}_{i,j} = \{\boldsymbol{h}_T^{i,j,m}\}_{m=1}^M$ consists of $M$ optimization histories $\boldsymbol{h}_T^{i,j,m} = \{(\boldsymbol{x}_t, y_t)\}_{t=1}^T$, where $\boldsymbol{x}_t$ is the query point selected by $\mathcal{A}_j$ at iteration $t$, and $y_t = f_i(\boldsymbol{x}_t)$ is its objective value. If the context is clear, we will omit $i, j, m$ and simply use $\boldsymbol{h}_T$ to denote a history with length $T$. The initial history $\boldsymbol{h}_0$ is defined as $\emptyset$. We impose no additional assumptions about the behavior algorithms, allowing for a range of BBO algorithms, even random search.

With the datasets, we seek to learn a model $\mathcal{M}_{\boldsymbol{\theta}}(\boldsymbol{x}_t|\boldsymbol{h}_{t-1})$, which is parameterized by $\boldsymbol{\theta}$ and generates the next query point $\boldsymbol{x}_t$ by conditioning on the previous history $\boldsymbol{h}_{t-1}$. As introduced in

Section 2.1, with a given budget $T$ and the history $\boldsymbol{h}_T$ produced by an algorithm $\mathcal{A}$, we use the cumulative regret

$$\text{Reg}_T = \sum_{t=1}^{T} (y^* - y_t) \tag{1}$$

as the evaluation metric, where $y^*$ is the optimum value and $\{y_t\}_{t=1}^{T}$ are the function values in $\boldsymbol{h}_T$.

## 3.1 METHOD OUTLINE

Given the current history $\boldsymbol{h}_{t-1}$ at iteration $t$, a BBO algorithm selects the next query point $\boldsymbol{x}_t$, observes the function value $y_t = f_i(\boldsymbol{x}_t)$, and updates the history $\boldsymbol{h}_t = \boldsymbol{h}_{t-1} \cup \{(\boldsymbol{x}_t, y_t)\}$. Similar to the previous work Chen et al. (2017), we take this framework as a starting point and treat the learning of a universal BBO algorithm as learning a model $\mathcal{M}_{\boldsymbol{\theta}}$, which takes the preceding history $\boldsymbol{h}_{t-1}$ as input and outputs a distribution of the next query point $\boldsymbol{x}_t$. The optimization histories in offline datasets provide a natural supervision for the learning process.

Suppose we have a set of histories $\{\boldsymbol{h}_T\}$, generated by a single behavior algorithm $\mathcal{A}$ on a single task $f$. By employing a causal transformer model $\mathcal{M}_{\boldsymbol{\theta}}$, we expect $\mathcal{M}_{\boldsymbol{\theta}}$ to imitate $\mathcal{A}$ and produce similar optimization history on $f$. In practice, we usually have datasets containing histories from multiple behavior algorithms $\{\mathcal{A}_j\}_{j=1}^{K}$ on multiple tasks $\{f_i\}_{i=1}^{N}$. To fit $\mathcal{M}_{\boldsymbol{\theta}}$, we use the negative log-likelihood loss

$$\mathcal{L}_{\text{BC}}(\boldsymbol{\theta}) = -\mathbb{E}_{\boldsymbol{h}_T \sim \mathcal{D}_{i,j}} \left[ \sum_{t=1}^{T} \log \mathcal{M}_{\boldsymbol{\theta}}(\boldsymbol{x}_t | \boldsymbol{h}_{t-1}) \right]. \tag{2}$$

To effectively minimize the loss, $\mathcal{M}_{\boldsymbol{\theta}}$ needs to recognize both the task and the behavior algorithm in-context, and then imitate the optimization behavior of the corresponding behavior algorithm.

Nevertheless, naively imitating the offline datasets hinders the model since some inferior behavior algorithms may severely degenerate the model's performance. Inspired by DT Chen et al. (2021), we propose to augment the optimization history with Regret-To-Go (RTG) tokens $R_t$, defined as the sum of instantaneous regrets over the future history:

$$\hat{\boldsymbol{h}}_T = \{(\boldsymbol{x}_t, y_t, R_t)\}_{t=0}^{T}, \ R_t = \sum_{t'=t+1}^{T} (y^* - y_{t'}), \tag{3}$$

where $\boldsymbol{x}_0$ and $y_0$ are placeholders for padding, denoted as [PAD] in Figure 1(b), and $R_T = 0$. The augmented histories compose the augmented dataset $\widehat{\mathcal{D}}_{i,j}$, and the training objective of $\mathcal{M}_{\boldsymbol{\theta}}$ becomes

$$\mathcal{L}_{\text{RIBBO}}(\boldsymbol{\theta}) = -\mathbb{E}_{\hat{\boldsymbol{h}}_T \sim \widehat{\mathcal{D}}_{i,j}} \left[ \sum_{t=1}^{T} \log \mathcal{M}_{\boldsymbol{\theta}}(\boldsymbol{x}_t | \hat{\boldsymbol{h}}_{t-1}) \right]. \tag{4}$$

The integration of RTG tokens in the context brings identifiability of behavior algorithms, and the model $\mathcal{M}_{\boldsymbol{\theta}}$ can effectively utilize them to make appropriate decisions. Furthermore, RTG tokens have a direct correlation with the metric of interest, i.e., cumulative regret $\text{Reg}_T$ in Eq. (1). Conditioning on a lower RTG token provides a guidance to our model and reinforces $\mathcal{M}_{\boldsymbol{\theta}}$ to exhibit superior performance. These advantages will be clearly shown by experiments in Section 4.4.

The resulting method RIBBO has implicitly utilized the in-context learning capacity of transformer to guide the optimization with previous histories and the desired future regret as context. The in-context learning capacity of inferring the tasks at hand based on the input contexts has been observed as the scale of data and model increases Kaplan et al. (2020). It has been explored to infer general functional relationships as supervised learning or RL algorithms. For example, the model is expected to predict accurately on the query input $\boldsymbol{x}_t$ by feeding the training dataset $\{(\boldsymbol{x}_i, y_i)\}_{i=1}^{t-1}$ as the context Guo et al. (2023); Hollmann et al. (2023); Li et al. (2023); Laskin et al. (2023) learned RL algorithms using causal transformers. Here, we use it for BBO.

## 3.2 PRACTICAL IMPLEMENTATION

Next, we detail the model architecture, training, and inference of RIBBO.

**Model Architecture.** For the formalization of the model $\mathcal{M}_{\boldsymbol{\theta}}$, we adopt the commonly used GPT architecture Radford et al. (2018), which comprises a stack of causal attention blocks. Each block

---

**Algorithm 1** Model Inference with HRR

---

**Input**: trained model $\mathcal{M}_{\boldsymbol{\theta}}$, budget $T$, optimum value $y^*$
**Process**:

1: Initialize $\hat{\boldsymbol{h}}_0 = \{(\boldsymbol{x}_0, y_0, R_0)\}$, where $\boldsymbol{x}_0$ and $y_0$ are placeholders for padding and $R_0 = 0$;
2: **for** $t = 1, 2, \ldots, T$ **do**
3:     Generate the next query point $\boldsymbol{x}_t \sim \mathcal{M}_{\boldsymbol{\theta}}(\cdot | \hat{\boldsymbol{h}}_{t-1})$;
4:     Evaluate $\boldsymbol{x}_t$ to obtain $y_t = f(\boldsymbol{x}_t)$;
5:     Calculate the instantaneous regret $r = y^* - y_t$;
6:     Relabel $R_i \leftarrow R_i + r$, for each $(\boldsymbol{x}_i, y_i, R_i)$ in $\hat{\boldsymbol{h}}_{t-1}$;
7:     $\hat{\boldsymbol{h}}_t = \hat{\boldsymbol{h}}_{t-1} \cup \{(\boldsymbol{x}_t, y_t, 0)\}$
8: **end for**

---

is composed of an attention mechanism and a feed-forward network. We aggregate each triplet $(\boldsymbol{x}_i, y_i, R_i)$ using a two-layer MLP network. The output of $\mathcal{M}_{\boldsymbol{\theta}}$ is a diagonal Gaussian distribution of the next query point. Note that previous works that adopt the sequence model as surrogate models Müller et al. (2022; 2023); Nguyen & Grover (2022) typically remove the positional encoding because the surrogate model should be invariant to the history order. On the contrary, our implementation preserves the positional encoding, naturally following the behavior of certain algorithms (e.g., BO or EA) and making it easier to learn from algorithms. Additionally, the positional encoding can help maintain the monotonically decreasing order of RTG tokens. More details about the architecture can be found in Appendix A.

**Model Training.** RTG tokens are calculated as outlined in Eq. (3) for the offline datasets before training. Since the calculation of regret requires the optimum value of task $i$, we use the best-observed value $y^i_{\max}$ as a proxy for the optimum. Let $\{\widehat{\mathcal{D}}_{i,j}\}_{i \in [N], j \in [K]}$ denote the RTG augmented datasets with $N$ tasks and $K$ algorithms. During training, we sample a minibatch of consecutive subsequences of length $\tau < T$ uniformly from the augmented datasets. The training objective is to minimize the RIBBO loss in Eq. (4).

**Model Inference.** The model $\mathcal{M}_{\boldsymbol{\theta}}$ generates the query points $\boldsymbol{x}_t$ auto-regressively during inference, which involves iteratively selecting a new query point $\boldsymbol{x}_t$ based on the current augmented history $\hat{\boldsymbol{h}}_{t-1}$, evaluating the query as $y_t = f(\boldsymbol{x}_t)$, and updating the history by $\hat{\boldsymbol{h}}_t = \hat{\boldsymbol{h}}_{t-1} \cup \{(\boldsymbol{x}_t, y_t, R_t)\}$. A critical aspect of this process is how to specify the value of RTG (i.e., $R_t$) at every iteration $t$. Inspired by DT, a naive approach is to specify a desired performance as the initial RTG $R_0$, and decrease it as $R_t = R_{t-1} - (y^* - y_t)$. However, this strategy has the risk of producing out-of-distribution RTGs, since the values can fall below $0$ due to an improperly selected $R_0$.

Given the fact that RTGs are lower bounded by $0$ and a value of $0$ implies a good BBO algorithm with low regret, we propose to set the immediate RTG as $0$. Furthermore, we introduce a strategy called **Hindsight Regret Relabelling (HRR)** to update previous RTGs based on the current sample evaluations. The inference procedure with HRR is detailed in Algorithm 1. In line 1, the history $\hat{\boldsymbol{h}}_0$ is initialized with padding placeholders $\boldsymbol{x}_0, y_0$ and RTG $R_0 = 0$. At iteration $t$ (i.e., lines 3–7), the model $\mathcal{M}_{\boldsymbol{\theta}}$ is fed with the augmented history $\hat{\boldsymbol{h}}_{t-1}$ to generate the next query point $\boldsymbol{x}_t$ in line 3, followed by the evaluation procedure to obtain $y_t$ in line 4. Then, the immediate RTG $R_t$ is set to $0$, and we employ HRR to update previous RTG tokens in $\hat{\boldsymbol{h}}_{t-1}$, i.e., calculate the instantaneous regret $r = y^* - y_t$ (line 5) and add $r$ to every RTG token within $\hat{\boldsymbol{h}}_{t-1}$ (line 6):

$$\forall 0 \leq i < t, R_i \leftarrow R_i + (y^* - y_t). \tag{5}$$

Note that this relabelling process guarantees that $\forall 0 \leq i < t$, the RTG token $R_i = \sum_{t'=i+1}^{t}(y^* - y_{t'})$, which can also be written as $\sum_{t'=i+1}^{T}(y^* - y_{t'})$, consistent with the definition in Eq. (3), because the immediate RTG $R_t = \sum_{t'=t+1}^{T}(y^* - y_{t'})$ is set to $0$. In line 7, the history $\hat{\boldsymbol{h}}_t$ is updated by expanding $\hat{\boldsymbol{h}}_{t-1}$ with $\{(\boldsymbol{x}_t, y_t, 0)\}$, i.e., the current sampling and its immediate RTG $R_t = 0$. The above process is repeated until reaching the budget $T$. Thus, we can find that HRR not only exploits the full potential of $\mathcal{M}_{\boldsymbol{\theta}}$ through using $0$ as the immediate RTG and thereby demands the model to generate the most advantageous decisions, but also preserves the calculation of RTG tokens following the same way as the training data, i.e., representing the cumulative regret over future optimization history.

### 3.3 DATA GENERATION

Finally, we give some guidelines about data generation for using the proposed RIBBO method.

**Data Collection.** Given a set of tasks $\{f_i\}_{i=1}^N$ sampled from the task distribution $P(\mathcal{F})$, we can employ a diverse set of behavior algorithms for data collection. For example, we can select some representatives from different types of BBO algorithms, e.g., BO and EA. Datasets $\mathcal{D}_{i,j}$ are obtained by using each behavior algorithm to optimize each task with different random seeds. Each optimization history $\boldsymbol{h}_T = \{(\boldsymbol{x}_t, y_t)\}_{t=1}^T$ in $\mathcal{D}_{i,j}$ is then augmented with RTG tokens $R_t$, which is computed as in Eq. (3). The resulting histories $\hat{\boldsymbol{h}}_T = \{(\boldsymbol{x}_t, y_t, R_t)\}_{t=0}^T$ compose the final datasets $\widehat{\mathcal{D}}_{i,j}$ for model training.

**Data Normalization.** To provide a unified interface and balance the statistic scales across tasks, it is important to apply normalization to the inputs to our model. We normalize the point $\boldsymbol{x}$ by $(\boldsymbol{x} - \boldsymbol{x}_{\min})/(\boldsymbol{x}_{\max} - \boldsymbol{x}_{\min})$, with $\boldsymbol{x}_{\max}$ and $\boldsymbol{x}_{\min}$ being the upper and lower bounds of the search space, respectively. For the function value $y$, we apply random scaling akin to previous works Wistuba & Grabocka (2021); Chen et al. (2022). That is, when sampling a history $\boldsymbol{h}_\tau$ from the datasets $\mathcal{D}_{i,j}$, we randomly sample the lower bound $l \sim \mathcal{U}(y_{\min}^i - \frac{s}{2}, y_{\min}^i + \frac{s}{2})$ and the upper bound $u \sim \mathcal{U}(y_{\max}^i - \frac{s}{2}, y_{\max}^i + \frac{s}{2})$, where $\mathcal{U}$ stands for uniform distribution, $y_{\min}^i, y_{\max}^i$ denote the observed minimum and maximum values for $f_i$, and $s = y_{\max}^i - y_{\min}^i$; the values $y_t$ in $\boldsymbol{h}_\tau$ are then normalized by $(y_t - l)/(u - l)$ for training. The RTG tokens are calculated accordingly with the normalized values. The random normalization can make a model exhibit invariance across various scales of $y$. For inference, the average values of the best-observed and worst-observed values across the training tasks are used to normalize $y$.

## 4 EXPERIMENTS

In this section, we examine the performance of RIBBO on a wide range of tasks, including synthetic functions, Hyper-Parameter Optimization (HPO) and robot control problems. The model architecture and hyper-parameters are maintained consistently across these problems. We train our model using five distinct random seeds, ranging from $0$ to $4$, and each trained model is run five times independently during the execution phase. We will report the average performance and standard deviation. Details of the model hyper-parameters are given in Appendix A. The codes are provided in the supplementary.

### 4.1 EXPERIMENTAL SETUP

**Benchmarks.** We use BBO Benchmarks BBOB Elhara et al. (2019), HPO-B Arango et al. (2021), and rover trajectory planning task Wang et al. (2018). The BBOB suite, a comprehensive and widely used benchmark in the continuous domain, consists of $24$ synthetic functions. For each function, a series of linear and non-linear transformations are implemented on the search space to obtain a distribution of functions with similar properties. According to the properties of these functions, they can be divided into $5$ categories, and we select one from each category due to resource constraints, including Greiwank Rosenbrock, Lunacek, Rastrigin, Rosenbrock, and Sharp Ridge. HPO-B is a commonly used HPO benchmark and consists of a series of HPO problems. Each problem is to optimize a machine learning model across various datasets, and an XGBoost model is provided as the objective function for evaluation in a continuous space. We conduct experiments on two widely used models, SVM and XGBoost, in the continue domain. For robot control optimization, we perform experiments on rover trajectory planning task, which is a trajectory optimization problem to emulate rover navigation. Similar to Elhara et al. (2019); Volpp et al. (2020), we implement random translations and scalings to the search space to construct a distribution of functions. For BBOB and rover problems, we sample a set of functions from the task distribution as training and test tasks, while for HPO-B, we use the meta-training/test task splits provided by the authors. Detailed explanations of the benchmarks can be found in Appendix B.1.

**Data.** Similar to OptFormer Chen et al. (2022), we employ 7 behavior algorithms, i.e., Random Search, Shuffled Grid Search, Hill Climbing, Regularized Evolution Real et al. (2019), Eagle Strategy Yang & Deb (2010), CMA-ES Hansen (2016), and GP-EI Balandat et al. (2020), which are representatives of heuristic search, EA, and BO, respectively. Datasets are generated by employing

each behavior algorithm to optimize various training functions sampled from the task distribution, using different random seeds. For inference, new test functions are sampled from the same task distribution to serve as the test set. Specifically, for the HPO-B problem, the meta-training/test splits have been predefined by the authors and we adhere to this standard setup. Additional information about the behavior algorithms and datasets can be found in Appendix B.2 and B.3.

## 4.2 BASELINES

As RIBBO is an in-context E2E model, the most related baselines are those also training an E2E model with offline datasets, including Behavior Cloning (BC) Bain & Sammut (1995) and Opt-Former Chen et al. (2022). Their hyper-parameters are set as same as that of our model for fairness. Note that the seven behavior algorithms used to generate datasets are also important baselines, and included for comparison as well.

**BC** uses the same transformer architecture as RIBBO. The only difference is that we do not feed RTG tokens into the model of BC and train to minimize the BC loss in Eq. (2). When the solutions are generated auto-regressively, BC tends to imitate the average behavior of various behavior algorithms. Consequently, the inclusion of underperforming algorithms, e.g., Random Search and Shuffled Grid Search, may significantly degrade the performance. To mitigate this issue, we have also trained the model by excluding these underperforming algorithms, denoted as **BC Filter**.

**OptFormer** employs a transformer to imitate the behaviors of a set of algorithms and an algorithm identifier usually needs to be specified manually during inference for superior performance. Its original implementation is built upon a text-based transformer with a large training scale. In this paper, we re-implement a simplified version of OptFormer where we only retain the algorithm identifier within the metadata. The initial states, denoted as $x_0$ and $y_0$, are used to distinguish between algorithms. They are obtained by indexing the algorithm type through an embedding layer, thereby aligning the initial states with the specific imitated algorithm. This enables the identification of distinct behavior algorithms within the simplified OptFormer. Further details about the re-implementation can be found in Appendix C.

## 4.3 MAIN RESULTS

The results are shown in Figure 2. For the sake of clarity in visualization, we have omitted the inclusion of Random Search and Shuffled Grid Search due to their poor performance from start to finish. We can observe that RIBBO achieves superior or at least equivalent efficacy in comparison to the best behavior algorithm on each problem except SVM and rover. This demonstrates the versatility of RIBBO, while the most effective behavior algorithm depends upon the specific problem at hand, e.g., the best behavior algorithms on Lunacek, Rastrigin and XGBoost are GP-EI, Eagle Strategy and CMA-ES, respectively. Note that the good performance of RIBBO does not owe to the memorization of optimal solutions, as the search space is transformed randomly, resulting in variations in optimal solutions across different functions from the same distribution. It is because RIBBO is capable of using RTG tokens to identify algorithms and reinforce the performance on top of the behavior algorithms, which will be clearly shown later. We can also observe that RIBBO performs extremely well in the early stage, which draws the advantage from the HRR strategy, i.e., employing 0 as the immediate RTG to generate the optimal potential solutions.

RIBBO does not perform well on the SVM problem, which may be due to the problem's low-dimensional nature (only three parameters) and its relative simplicity for optimization. Behavior algorithms can achieve good performance easily, while the complexity of RIBBO's training and inference processes could instead result in the performance degradation. For the rover problem where GP-EI performs the best, we collect less data from GP-EI than other behavior algorithms due to the high time cost. This may limit RIBBO's capacity to leverage the high-quality data from GP-EI, given its small proportion relative to the data collected from other behavior algorithms. Despite this, RIBBO is still the runner-up, significantly surpassing the other behavior algorithms.

Compared with BC and BC Filter, RIBBO performs consistently better except on the SVM problem. BC tends to imitate the average behavior of various algorithms, and its poor performance is due to the aggregation of behavior algorithms with inferior performance. BC Filter is generally better than BC, because the data from the two underperforming behavior algorithms, i.e., Random

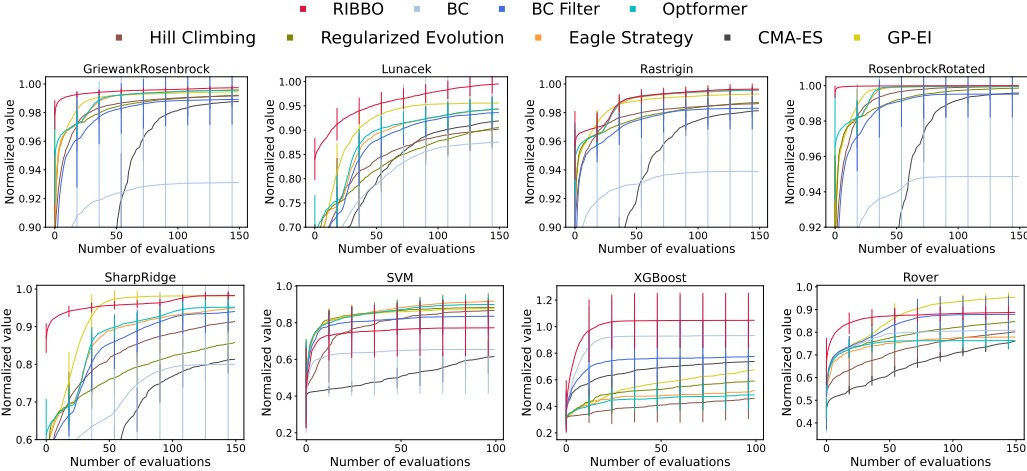

Figure 2: Performance comparison among RIBBO, BC, BC Filter, OptFormer, and behavior algorithms on synthetic functions, HPO, and robot control problems. The $y$-axis is the normalized average objective value, and the length of vertical bars represents the standard deviation.

Search and Shuffled Grid Search, are manually excluded for the training of BC Filter. As introduced before, OptFormer requires to manually specify which algorithm to execute. We have specified the behavior algorithm Eagle Strategy in Figure 2, which obtains good overall performance on these problems. It can be observed that OptFormer displays a close performance to Eagle Strategy, while RIBBO performs better. More results about the imitation capacity of OptFormer can be found in Appendix C.

**Why Does RIBBO Behave Well?** To better understand RIBBO, we train the model using only two behavior algorithms, Eagle Strategy and Random Search, which represent a good algorithm and an underperforming one, respectively. Figure 3(a) visualizes the contour lines of the 2D Branin function and the sampling points of RIBBO, Eagle Strategy, and Random Search, represented by red, orange, and gray points, respectively. The arrows are used to represent the optimization trajectory of RIBBO. Note that the two parameters of Branin have been scaled to $[-1, 1]$ for better visualization. It can be observed that RIBBO makes a prediction preferring Eagle Strategy over Random Search, indicating its capability to automatically identify the quality of training data. Additionally, RIBBO achieves the exploration and exploitation trade-off capability using its knowledge about the task obtained during training, thus generating superior solutions over the ones in the training dataset.

**Generalization.** We also conduct experiments to examine the generalization capabilities of RIBBO. For this purpose, we train the model on all 24 BBOB synthetic functions simultaneously with the results shown in Figure 3(b). To aggregate results across functions with different output scaling, we normalize all the functions adhering to previous literature Turner et al. (2020); Arango et al. (2021); Chen et al. (2022). The results suggest that RIBBO demonstrates strong generalizing to a variety of functions with different properties. Further experiments are conducted to examine the cross-distribution generalization to unseen function distributions. The model is trained on 4 of the 5 chosen synthetic functions and tested with the remaining one. Note that each function (i.e., Greiwank Rosenbrock, Lunacek, Rastrigin, Rosenbrock, and Sharp Ridge) here actually represents a distribution of functions with similar properties, and a set of functions is sampled from each distribution as introduced before. The results suggest that RIBBO has a strong generalizing ability to unseen function distributions. Due to the space limitation, the results and more details on cross-distribution generalization are deferred to Appendix D. The good generalization of RIBBO can be attributed to the paradigm of learning the entire algorithm, which can acquire general knowledge, such as exploration and exploitation trade-off from data, as observed in Chen et al. (2017). In contrast, such generalization may be limited if we learn surrogate models from data, because the function landscape inherent to surrogate models will contain only the knowledge of similar functions.

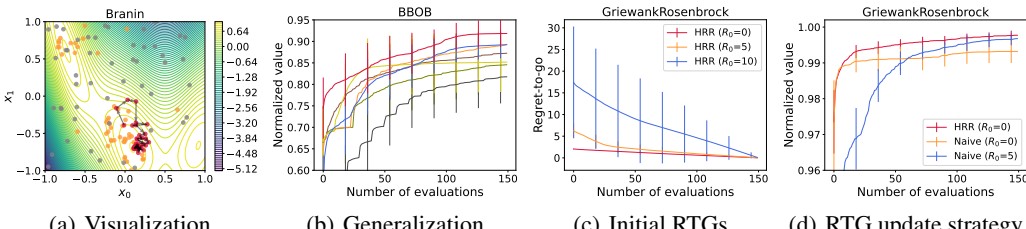

|  (a) Visualization | (b) Generalization | (c) Initial RTGs | (d) RTG update strategy |

Figure 3: *(a) Visualization* of the contour lines of 2D Branin function and sampling points of RIBBO (red), Eagle Strategy (orange), and Random Search (gray), where the arrows represent the optimization trajectory of RIBBO. *(b) Generalization* by training the model across all $24$ BBOB synthetic functions simultaneously. The results across functions with different output scaling are normalized to obtain the aggregate results. The legend shares with that of Figure 2. *(c) Initial RTG $R_0$*'s influence on performance. *(d) RTG update strategy* comparison between HRR and the naive strategy with various initial RTG $R_0$.

## 4.4 Ablation Studies

RIBBO augments the histories with RTG tokens, facilitating distinguishing algorithms and automatically generating algorithms with user-specified performance. Next, we will verify the effectiveness of RTG conditioning and HRR strategy.

**Influence of Initial RTG Token $R_0$.** By incorporating RTG tokens, RIBBO is able to attend to RTGs and generate optimization trajectories based on the specified initial RTG token $R_0$. To validate this, we examine the performance of RIBBO with different values of $R_0$, and the results are presented in Figure 3(c). Here, the RTG values, instead of normalized objective values, are used as the $y$-axis. We can observe that the cumulative regrets of the generated query sequence do correlate with the specified RTG, indicating that RIBBO establishes the connection between regret and generation. We also conduct experiments to explore the effect of varying the immediate RTG $R_t$, and it is observed that setting to a value larger than $0$ will decrease the performance and converge to a worse value. Please see Appendix E.

**Effectiveness of HRR.** A key point of the inference procedure is how to update the value of RTG tokens at each iteration. To assess the effectiveness of the proposed strategy HRR outlined in Eq. (5), we compare it with the naive strategy, that sets an initial RTG token $R_0$ and decreases it by the one-step regret after each iteration, a method that employs the same updating strategy mechanism as DT. The results are shown in Figure 3(d). The naive strategy displays distinct behaviors depending on the initial setting of $R_0$. Specifically, when $R_0 = 0$, i.e., the lower bound of regret, the model performs well initially. However, as the optimization progresses, the RTG tokens gradually decrease to negative values, leading to poor performance since negative RTGs are out-of-distribution values. Using $R_0 = 5$ compromises the initial performance, as the model may not select the most aggressive solutions with a high $R_0$. However, a higher initial $R_0$ yields better convergence value since it prevents out-of-distribution RTGs in later stage. The proposed HRR strategy consistently outperforms across the whole optimization stage, because setting the immediate RTG to $0$ encourages the model to make the most advantageous decisions at every iteration, while hindsight relabeling of previous RTG tokens, as specified in Eq. (5), ensures that these values remain meaningful and feasible.

**Further Studies.** We also study the effects of the method to aggregate $(\boldsymbol{x}_i, y_i, R_i)$ tokens, the normalization method for $y$, the model size, and the sampled subsequence length $\tau$. Please see Appendix F. More visualizations illustrating the effects of random transformations on the search space are detailed in Appendix G.

## 5 Conclusion

This paper proposes RIBBO, which employs a transformer architecture to learn a reinforced BBO algorithm from offline datasets in an E2E fashion. By incorporating RTG tokens into the optimization histories, RIBBO can automatically generate optimization trajectories satisfying the user-desired regret. Comprehensive experiments on BBOB synthetic functions, HPO and robot control problems show the versatility of RIBBO.

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

## A    MODEL DETAILS

We employ the commonly used GPT architecture Radford et al. (2018) and the hyper-parameters are maintained consistently across the problems. Details can be found in Table 1. For BC, BC Filter and OptFormer, the hyper-parameters are set as same as those of our model. The training takes about 20 hours on 1 GPU (Nvidia RTX 4090).

## B    DETAILS OF EXPERIMENTAL SETUP

### B.1    BENCHMARKS

- **BBOB** Elhara et al. (2019) is a widely used synthetic BBO benchmark, consisting of 24 synthetic functions in the continuous domain. This benchmark makes a series of transformations in the search space, such as linear transformations (e.g., translation, rotation, scaling) and non-linear transformations (e.g., Tosz, Tasy), to create a distribution of functions while retaining similar properties. According to the properties of functions, these synthetic functions can be divided into 5 categories, i.e., (1) separable functions, (2) moderately conditioned functions, (3) ill-conditioned and unimodal functions, (4) multi-modal functions with adequate global structure, and (5) multi-modal functions with weak global structures. We select one function from each category to evaluate our algorithm, Rastrigin

Table 1: List of hyper-parameter settings in RIBBO.

| RIBBO | |
|---|---|
| Embedding dimension | 256 |
| Number of self-attention layers | 12 |
| Number of self-attention heads | 8 |
| Point-wise feed-forward dimension | 1024 |
| Dropout rate | 0.1 |
| Batch size | 64 |
| Learning rate | 0.0002 |
| Learning rate decay | 0.01 |
| Optimizer | Adam |
| Optimizer scheduler | Linear warm up and cosine annealing |
| Number of training steps | 500,000 |
| Length of subsequence $\tau$ | 50 |

from (1), Rosenbrock Rotated from (2), Sharp Ridge from (3), Greiwank Rosenbrock from (4), and Lunacek from (5). We use the BBOB benchmark implementation in Open Source Vizier[1], and the dimension is set to 10 for all functions.

- **SVM and XGBoost.** HPO-B Arango et al. (2021) is the most commonly used HPO benchmark and is grouped by search space id. Each search space id corresponds to a machine learning model, e.g., SVM or XGBoost. Each such search space has multiple associated dataset id, which is a particular HPO problem, i.e., optimizing the performance of the corresponding model on a dataset. For the continuous domain, it fits an XGBoost model as the objective function for each HPO problem. These datasets for each search space id are divided into training and test datasets. We examine our method on two selected search space id, i.e., 5527 and 6767, which are two representative HPO problems tuning SVM and XGBoost, respectively. SVM has 3 parameters, while XGBoost has 18 parameters, which is the most in HPO-B. We use the official implementations[2].

- **Rover Trajectory Planning** Eriksson et al. (2019); Wang et al. (2018) is a trajectory optimization task designed to emulate a rover navigation task. The trajectory is determined by fitting a B-spline to 30 points in a 2D plane, resulting in a total of 60 parameters to optimize. Given $\boldsymbol{x} \in [0, 1]^{60}$, the objective function is $f(\boldsymbol{x}) = c(\boldsymbol{x}) + \lambda(\|\boldsymbol{x}_{0,1} - \boldsymbol{s}\|_1 + \|\boldsymbol{x}_{58,59} - \boldsymbol{g}\|_1) + b$, where $c(\boldsymbol{x})$ is the cost of the given trajectory, $\boldsymbol{x}_{0,1}$ denotes the first and second dimensions of $\boldsymbol{x}$ (similarly for $\boldsymbol{x}_{58,59}$), $\boldsymbol{s}$ and $\boldsymbol{g}$ are 2D points that specify the starting and goal positions in the plane, $\lambda$ and $b$ are parameters to define the problem. This problem is non-smooth, discontinuous, and concave over the first two and last two dimensions. To construct the distribution of functions, we applied translations in $[-0.1, 0.1]^d$ and scalings in $[0.9, 1.1]$, similar to previous works Elhara et al. (2019); Volpp et al. (2020). The training and test tasks are randomly sampled from the distribution. We use the standard implementation for the rover problem[3].

## B.2 BEHAVIOR ALGORITHMS

The datasets are generated with several representatives of heuristic search, EA, and BO as behavior algorithms. We use the implementation in Open Source Vizier Song et al. (2022) for Random Search, Shuffled Grid Search, Eagle Strategy, CMA-ES. For Hill Climbing and Regularized Evolution, we provide a simple re-implementation. We use the implementation in BoTorch Balandat et al. (2020) for GP-EI. The details of these behavior algorithms are summarized as follows.

- Random Search selects a point uniformly at random from the domain at each iteration.
- Shuffled Grid Search discretizes the ranges of real parameters into 100 equidistant points and selects a random point from the grid without replacement at each iteration.

---

[1]https://github.com/google/vizier
[2]https://github.com/releaunifreiburg/HPO-B
[3]https://github.com/uber-research/TuRBO

- Hill Climbing. At each iteration $t$, the current best solution $x_{best}$ is mutated (using the same mutation operation as Regularized Evolution) to generate $x_{next}$, which is then evaluated. If $f(x_{next}) > f(x_{best})$, $x_{best}$ is updated to $x_{next}$.

- Regularized Evolution Real et al. (2019) is an evolutionary algorithm with tournament selection and age-based replacement. We use a population size of 25 and a tournament size of 5. At each iteration, a tournament subset is randomly selected from the current population, and the solution with the maximum value is mutated. The mutation operation uniformly selects one of the parameters and mutates it to a random value within the domain.

- Eagle Strategy Yang & Deb (2010) without the Levy random walk, aka Firefly Algorithm, maintains a population of fireflies. Each firefly emits light whose intensity corresponds to the objective value. At each iteration, for each firefly, a weight is calculated to chase after a brighter firefly and actively move away from darker ones in its vicinity. The position is updated based on the calculated weight.

- CMA-ES Hansen (2016) is a popular evolutionary algorithm. At each iteration, candidate solutions are sampled from a multivariate normal distribution and evaluated, then the mean and covariance matrix are updated.

- GP-EI employs GP as the surrogate model and EI Jones et al. (1998) as the acquisition function for BO.

RIBBO is trained exclusively from offline datasets derived from various source tasks, which are sampled from the task distribution. Each dataset consists of several optimization histories, generated by running a behavior algorithm on a task. No additional assumptions are imposed to the data collection process or the behavior algorithms, thereby permitting the use of any behavior algorithm.

Nevertheless, we can choose appropriate behavior algorithms to help the training. The model is trained on the datasets collected by the behavior algorithms, and the characteristics of these datasets can influence the model's efficacy. It is advisable to employ well-performing and diverse behavior algorithms to create these datasets. If the datasets are from suboptimal behavior algorithms predominantly, there might be a decline in performance, even with the incorporation of RTG tokens. If the datasets are predominantly produced by behavior algorithms with homogeneous properties, the model may only be adept at addressing specific problems characterized by those properties. Conversely, if the behavior algorithms are diverse, the model can learn the strengths from various algorithms.

### B.3 Data

For BBOB and rover problem, a set of tasks is sampled from the task distribution and the above behavior algorithms are used to collect data. Specifically, for the BBOB suite, a total of 200 functions are sampled, and each behavior algorithm is executed 500 times, except for GP-EI, which is limited to 100 function samples due to its high time cost. The total number of optimization histories is about $100,000$ and the length of each history is 150. For the rover problem, 300 functions are sampled with each being run 500 times, whereas for GP-EI, a smaller number of 50 functions is selected, each being run 500 times. The total number of optimization histories is about $150,000$ and the length of each history is 150. For testing, we randomly sample 10 functions for each problem and run each algorithm multiple times to report the average performance and standard deviation.

For HPO-B, we use the "v3" meta-training/test splits provided by the authors, which consist of 51 training and 6 test tasks for SVM, and similarly, 52 training and 6 test tasks for XGBoost. All behavior algorithms employ 500 random seeds to collect the training datasets. The total number of optimization histories is about $25,000$ and the length of each history is 100.

## C Re-Implementation of OptFormer

OptFormer Chen et al. (2022) is a general optimization framework based on transformer Vaswani et al. (2017) and provides an interface to learn policy or function prediction. When provided with textual representations of historical data, OptFormer can determine the next query point $x_t$, acting as a policy. Additionally, if the context incorporate a possible query point $x_t$, the framework is capable

of predicting the corresponding $y_t$, thus serving as a prediction model. We focus on the aspect of policy learning, due to its greater relevance to our work.

The original implementation is built upon a text-based transformer and uses private datasets for training. In this paper, we have re-implemented a simplified version of OptFormer, where we omit the textual tokenization process and only retain the algorithm type within the metadata. Numerical inputs are fed into the model and we use different initial states $x_0$ and $y_0$ to distinguish among algorithms. The initial states are derived by indexing the algorithm type in an embedding layer, thereby enabling the identification of distinct behavior algorithms within OptFormer. The hyperparameters are set as same as our method.

To examine the algorithm imitation capability, we compare our re-implementation with the corresponding behavior algorithms. The results are shown in Figure 4. For clarity in the visual representation, we only plot a subset of the behavior algorithms, including Shuffled Grid Search, Hill Climbing, Regularized Evolution and Eagle Strategy. These behavior algorithms are plotted by solid lines, while their OptFormer counterparts are shown by dashed lines with the same color. Note that the figure uses the immediate function values as the $y$-axis to facilitate a comprehensive observation of the optimization process.

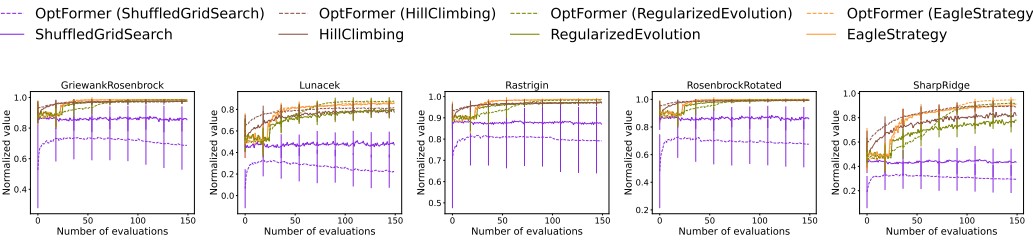

Figure 4: Comparison of the behavior algorithms with the OptFormer re-implementation.

## D GENERALIZATION

We conduct a series of experiments to examine the generalization of our method in this section. We train the model on $4$ of $5$ chosen synthetic functions and test on the remaining one. The results in Figure 5 have shown that RIBBO has a strong generalization ability to unseen function distributions.

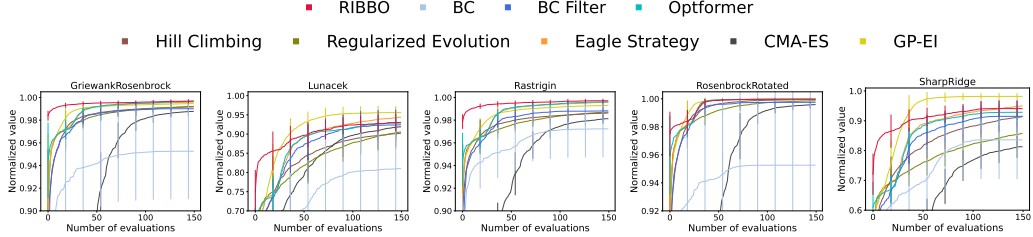

Figure 5: Cross-distribution generalization by training on $4$ of $5$ chosen synthetic functions and testing on the remaining one.

We also train the model across all training datasets from BBOB synthetic functions simultaneously, and normalize all the test functions to aggregate results across functions with different output scaling. The results are shown in Figure 6(a), and RIBBO has the best performance.

The generalization across different problems is implemented in an in-context manner. The context data, collected from the new problems, provides insights for understanding the problems at hand. These contexts are integrated to construct the inputs, thereby influencing the resulting sampled points.

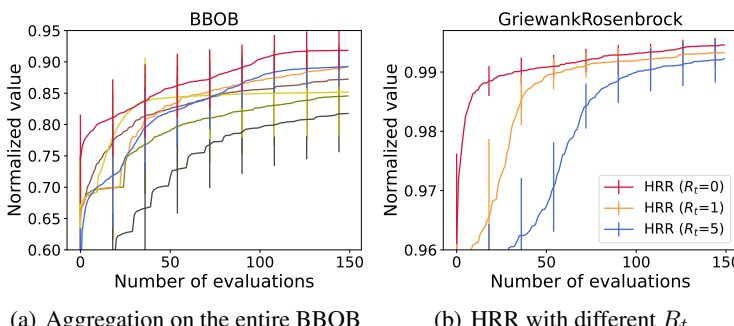

(a) Aggregation on the entire BBOB          (b) HRR with different $R_t$

# E   FURTHER DISCUSSIONS ABOUT RTG TOKENS

We conduct more discussion about the setting of RTG tokens in this section. The immediate RTG tokens are designed to represent the performance based on cumulative regret over the future trajectories (i.e., $R_i = \sum_{t'=i+1}^{t}(y^* - y_{t'})$), rather than focusing solely on instantaneous regret $y^* - y_{t+1}$. Thus, setting the immediate RTG to $0$ inherently accounts for future regret, thereby enabling the algorithm to autonomously trade-off the exploration and exploitation. We conduct experiments to examine the performance of RIBBO with different values of $R_t$. The results are shown in Figure 6(b). Setting to a value larger than $0$ will decrease the performance and converge to a worse value, validating the effectiveness of setting the immediate RTG as $0$.

Sampling the next query point $x_t$ is influenced by all the model inputs, including the immediate RTG token $x_t$ and the previous history $(x_i, y_i, R_i)_{i=0}^{t-1}$. The immediate RTG token represents the goal that is intended to be achieved, while the historical data contains information about the problem. These elements are integrated to construct the inputs, influencing the resulting sampled point. Even if the RTG token is set to $0$, implying a desire to sample the optimum, the short length of the history suggests limited knowledge about the problem, prompting the model to retain explorative behavior. As the history extends, suggesting a sufficient knowledge of the problem, the preference encoded by the RTG token shifts towards more greedy action. We conduct an experiment about the std of the output Gaussian head of $x$ of RIBBO using GriewankRosenbrock function and the results are shown in Table 2. It is clear that the std is very large in the early stage and becomes small later, which demonstrates the ability to trade-off the exploration and exploitation.

| Number of evaluations | 1 | 30 | 60 | 90 | 120 | 150 |
|---|---|---|---|---|---|---|
| std of $x$ | 0.37 | 0.17 | 0.14 | 0.11 | 0.03 | 0.01 |

Table 2: Std of the output Gaussian head of $x$.

# F   ABLATION STUDIES

We provide further ablation studies to examine the influence of the key components and hyperparameters in RIBBO, including the method to aggregate $(x_i, y_i, R_i)$ tokens, the normalization method for the function value $y$, the model size, and the sampled subsequence length $\tau$ during training.

**Token Aggregation** aims to aggregate the information from $(x_i, y_i, R_i)$ tokens and establish associations between them. RIBBO employs the Concatenation method (Concat), i.e., concatenating to aggregate $x_i$, $y_i$ and $R_i$ to form a single token. This method is compared with two alternative token aggregation methods, i.e., Addition (Add) and Interleaving (Interleave). The addition method integrates the values of each token into one, while the interleaving method addresses tokens sequentially. The results are shown in Figure 6(c). The concatenation method surpasses both the addition and interleaving methods, with the addition method showing the least efficacy. The concatenation method employs a relatively straightforward approach to aggregate tokens, while the interleaving method adopts a more complex process to learn the interrelations among tokens. The inferior performance

of the addition method can be due to the summation operator, which potentially eliminates critical details from the original tokens. These details are essential for generating the subsequent query $x_t$.

**Normalization Method** is to balance the scales of the function value $y$ across tasks. We compare the employed random normalization with the dataset normalization and the absence of any normalization. Dataset normalization adjusts the value of $y$ by $(y - y_{\min}^i)/(y_{\max}^i - y_{\min}^i)$ for each task $i$, where $y_{\min}^i$ and $y_{\max}^i$ denote the observed minimum and maximum values of $f_i$. No normalization uses the original function values directly. The results in Figure 6(d) indicate that random normalization and dataset normalization perform similarly, whereas the absence of normalization significantly hurts the performance. The choice of normalization method directly impacts the computation of RTG tokens and subsequently affects the generation of the desired regret. The lack of normalization leads to significant variations in the scale of $y$, which complicates the training process. Both random and dataset normalization scale the value of $y$ within a reasonable range, thus facilitating both training and inference. However, random normalization can bring additional benefits, such as invariance across various scales of $y$ as mentioned in Wistuba & Grabocka (2021) and Chen et al. (2022). Therefore, we recommend using random normalization in practice.

**Model Size** has an effect on the in-context learning ability. We assess the effects of different model sizes by comparing the performance of the currently employed model size with both a smaller and a larger model. The smaller model consists of 8 layers, 4 attention heads, and 128-dimensional embedding space, while the larger model has 16 layers, 12 attention heads, and 384-dimensional embedding space. The results are shown in Figure 6(e), indicating that the model size has a minimal impact on overall performance. Specifically, the smaller model, due to its limited capacity, shows a reduced performance, while the larger one, potentially more powerful, requires more training data, which can lead to a slight decrease in performance due to overfitting or inefficiency in learning from a limited dataset. This analysis highlights the trade-offs involved in selecting the appropriate model size for optimal performance.

**Subsequence Length** $\tau$ controls the context length during both the training and inference phases. The process of subsequence sampling acts as a form of data augmentation, enhancing training efficiency. As shown in Figure 6(f), sampling subsequences rather than using the entire history as context, particularly when $\tau = T = 150$, leads to improved performance. The computational complexity for causal transformer training and inference scales quadratically with the context length. Therefore, utilizing a shorter $\tau$ can significantly reduce computational demands. However, it is important to note that a shorter $\tau$ might not capture sufficient historical data, potentially degrading the performance due to the insufficient contextual information. This highlights the trade-offs between computational complexity and performance.

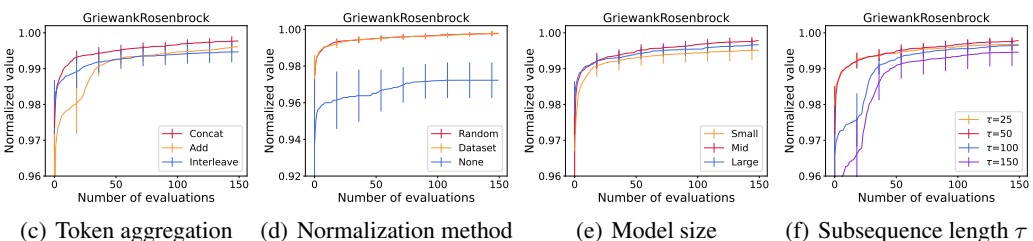

(c) Token aggregation    (d) Normalization method    (e) Model size    (f) Subsequence length $\tau$

Figure 6: Ablation studies of token aggregation, normalization method, model size, and subsequence length $\tau$.

# G    VISUALIZATION OF BRANIN FUNCTION

In several of our experiments, such as BBOB, rover problems, and the visualization analysis of the Branin function, we implement a series of transformations to the search space. This is designed to generate a distribution of functions with similar properties. A set of functions is sampled serving as the training and test tasks from the distribution.

For the Branin function, random translations and scalings are applied to form the distribution. In this section, we present visualizations of the contour lines of the 2D Branin functions, sampled from the

distribution to demonstrate the effects of the applied transformations. To this end, $4$ distinct random seeds are used to draw samples from the distribution. The visualizations are shown in Figure 7. Note that the two parameters of the Branin function have been scaled to the range of $[-1, 1]$ for the clarity of visual representations. Variations are observed in both the location of the optimum and the scales of the objective values. For instance, the optimum of the first subfigure locates more to the right compared to the second one, and the contour lines in the former are much more dispersed than those in the latter. The value scale for the first subfigure ranges from $-8$ to $1$, while in the second, it ranges from $-3.5$ to $1$. By applying these transformations, we can generate a function distribution that retains similar properties, enabling the sampling of training and test tasks for our model.

For other functions, such as BBOB suite, besides simple random translations and scalings, more complex transformations such as non-linear Tosz and Tasy transformations are applied, leading to a more intricate landscape. However, due to the high dimensionality of these functions, direct visualization is impractical, so we present only the visualization of the 2D Branin function.

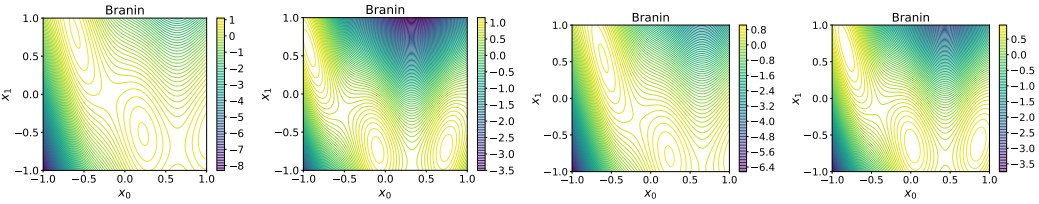

Figure 7: Contour line visualization for samples drawn from the 2D Branin function distribution using $4$ distinct random seeds.

## H  COMPARISON TO BBO ALGORITHM SELECTION METHODS AND META-LEARNING INDIVIDUAL COMPONENTS

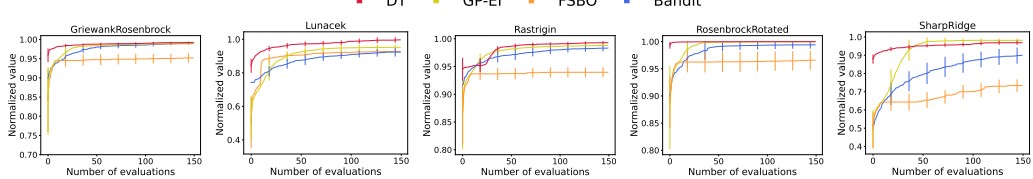

Figure 8: Comparison to BBO algorithm selection methods and meta-learning individual components.

## I  ADDITIONAL RESULTS ON BBOB FUNCTIONS

## J  FURTHER DISCUSSIONS ABOUT DT

DT represents a paradigm of upside down RL Schmidhuber (2019), that maps the desired rewards to corresponding actions to enhance the performance. The benefit of using regret-to-go is its independence from a predetermined horizon length when using the HRR strategy, contrasting with the return-to-go method where the calculation is usually related to the product of the maximal achievable reward and the horizon length. Once established, the return-to-go is expected to decrease as the optimization progresses. Additionally, from the perspective of optimization, the cumulative regret provides a more meaningful measure of the performance (measuring the gap to the optimum), compared to the sum of $y$. Therefore, regret-to-go has been adopted as the preferred metric in the algorithm's design.

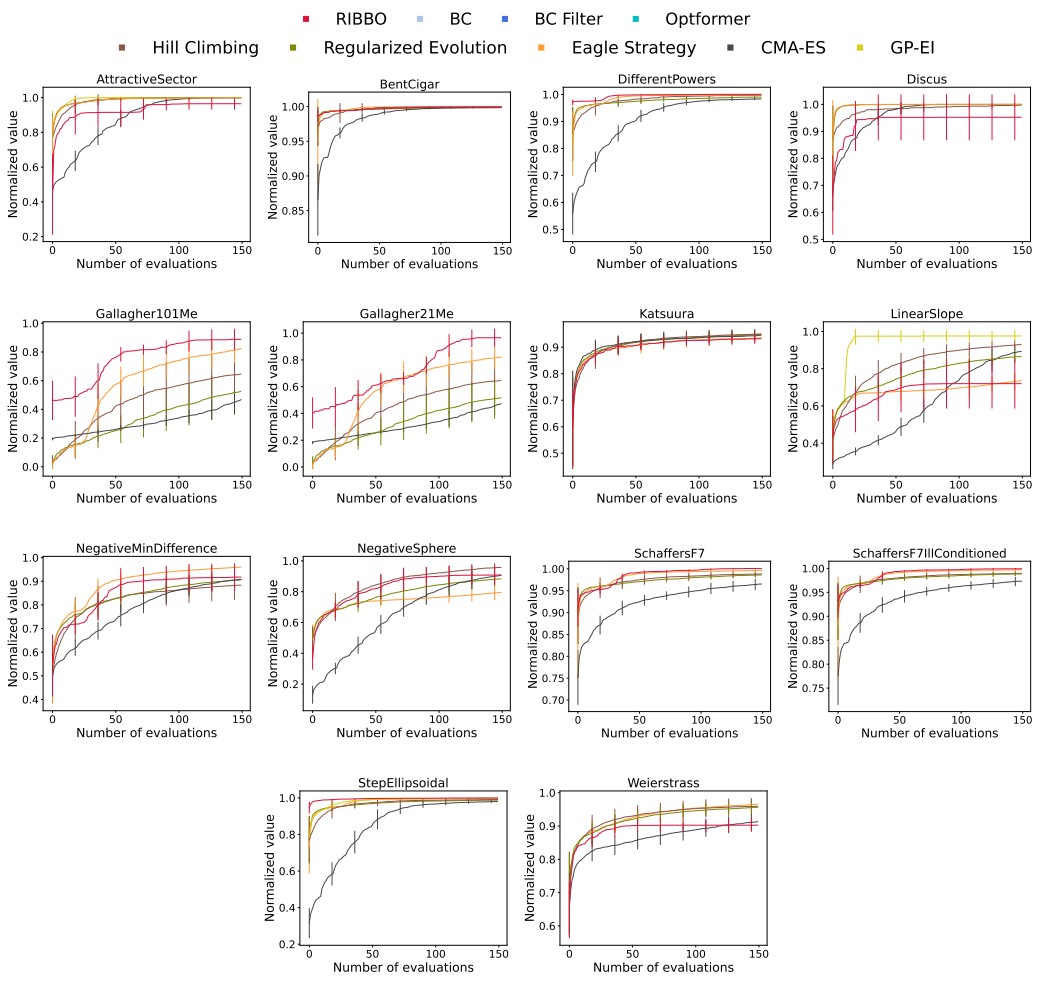

Figure 9: Additional results on BBOB functions.

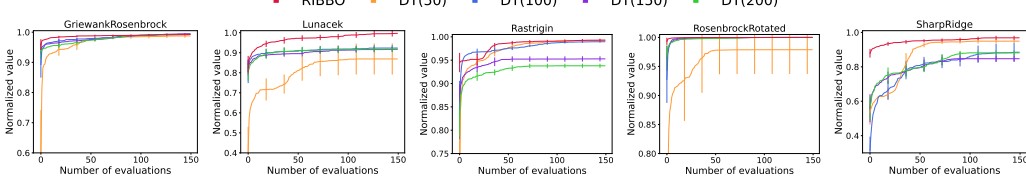

Figure 10: Comparison to DT with different initial return-to-go.

## K  FUTURE WORKS

In this paper, we only consider the model training over continuous search space with the same dimensionality, and it would be interesting to explore heteroscedastic search space with different types of variables. Usually, running BBO algorithms with more evaluations leads to improved performance. Training across diverse historical data lengths and incorporating length extrapolation techniques Press et al. (2022) to enable longer horizons may be helpful in more challenging scenarios. The OOD problems for RTG tokens are also important topic. Generalization across functions with disparate properties remains a significant challenge. Incorporating additional meta-features related to the problem, such as the range, type and statistical features of each variable may be important to develop a model that can recognize distinct problems and apply its learning to new problems,

thereby enhancing its generalization. A mathematical theoretical analysis about the cumulative regret analysis and generalization analysis based on RTG tokens is of interest and helpful for refining the algorithmic designs.

