# OpenReview forum: "Reinforced In-Context Black-Box Optimization"
_ICLR.cc/2025/Conference — Submitted to ICLR 2025_

### Official Review · Reviewer_dM2J · 2024-11-03

**Soundness:** 2
**Presentation:** 2
**Contribution:** 1
**Rating:** 3
**Confidence:** 3

**Summary:**

This paper presents a meta-learning algorithm designed for black-box optimization (BBO) problems, which are often derivative-free and lack a gradient signal. The proposed method uses an offline dataset collected from behavioral optimization algorithms, each of which generates sequences of optimization steps across various problems. The algorithm then trains a sequence-to-sequence model based on the Transformer architecture to replicate these sequences, aiming to generalize effectively and produce a high-quality optimization sequence when applied to new problems sampled from the training problem distribution.

In a typical optimization sequence, each step includes a pair of values, (x_i, y_i), where x and y represent the input and output of the BBO problem, respectively. The authors propose enhancing this dataset by adding a Regret-To-Go (RTG) scalar, represented as (x_i, y_i, r_i), where r_i is the sum of future regrets for t > i. This paper distinguishes between Behavioral Cloning (BC), which learns without RTG, and RIBBO, which incorporates RTG into the model training.

The authors test their algorithm across three datasets: (1) BBOB, a collection of synthetic optimization problems grouped by shared characteristics; (2) HPO-B, a set of machine learning hyperparameter optimization problems targeting base models such as XGBoost and SVM; and (3) Rover trajectory planning, where the goal is to optimize a sequence of 30 two-dimensional points that minimize a cost function. The algorithm is evaluated against other meta-learning BBO approaches, such as OptFormer and BC, as well as traditional BBO methods, including CMA-ES. RIBBO demonstrates strong performance, outperforming the alternatives in these experiments.

**Strengths:**

1. Data-Efficiency:

RIBBO addresses a significant problem in BBO by learning from sequences of sub-optimal trials. This approach enables the algorithm to learn from incomplete or unsuccessful trials, potentially reducing computational costs.

2. Implementation and Training Simplicity:

The method is relatively straightforward to implement and train, given the simplicity of the Transformer architecture and the offline training process.

**Weaknesses:**

1. Behavioral Policy Limitations:

The potential of RIBBO to outperform the behavioral policies it learns from is uncertain. Unlike reinforcement learning (RL), RIBBO lacks a policy optimization component; it merely learns to output the next query, x_i, based on past inputs, outputs, and RTG. This approach is essentially behavioral cloning that might favor sequences with lower regret if the RTG functions as intended. However, without optimizing the output sequences (e.g., by approximating the target output value, y, and deriving a gradient signal [1]), RIBBO’s ability to surpass the behavioral policies remains unclear.

2. Justification for RTG Inclusion:

The authors could strengthen their argument and visual representation of why RTG enhances solution quality. It is not evident why augmenting the input data with RTG would encourage the model to generate lower-regret sequences. Why is the network motivated to generate sequences of lower RTG?

3. Transferability Across BBO Problems:

Domains like RL or few-shot learning provide clear justifications for what knowledge can transfer across tasks, such as environment dynamics or domain distribution, respectively. However, in black-box problems, a shared generative function across tasks is not always present. Even in cases where tasks are sampled from a common distribution, such as in the same group in the BBOB testbench or the rover trajectory planning problem, RIBBO lacks an adaptation mechanism like MAML[2], which would help in adjusting to different environments. Consequently, the paper should address whether and why RIBBO is truly adaptable to unseen tasks.


References:
1. Sarafian, E., Sinay, M., Louzoun, Y., Agmon, N., & Kraus, S. (2020, July). Explicit Gradient Learning for Black-Box Optimization. In ICML (pp. 8480-8490).
2. Finn, Chelsea, Pieter Abbeel, and Sergey Levine. "Model-agnostic meta-learning for fast adaptation of deep networks." International conference on machine learning. PMLR, 2017.

**Questions:**

1. BBOB Experiment Clarifications:

In the BBOB experiments, what dimensions were selected, and what is the interpretation of the x-axis (number of evaluations)?

2. Hyperparameter Sensitivity:

A primary motivation for RIBBO is to alleviate the challenge of manually tuning hyperparameters (lines 13–16 in the abstract). However, RIBBO itself requires hyperparameter tuning due to the nature of neural network training. Some of these parameters are listed in Table 1 (Appendix). How were these hyperparameters tuned, and could the authors provide sensitivity metrics to show their impact on performance?

3. Literature Review Addition:

Opt-GAN [1] is another BBO method that aims to learn a global optimizer through a neural network, predicting the next candidate point based on previous points sampled in an off-policy fashion. It would be beneficial to include this work in the literature review.

References
1[] Lu, Minfang, et al. "OPT-GAN: a broad-spectrum global optimizer for black-box problems by learning distribution." Proceedings of the AAAI Conference on Artificial Intelligence, vol. 37, no. 10, 2023.

---

> ### Author Response · Authors · 2024-11-25
> **Response by the authors**
>
> Thanks for your valuable and constructive comments. Below please find our responses.
>
> ### Q1: Behavioral policy limitations
>
> Thanks for your comment. RIBBO is more close to the domain of offline RL [1], which leverages pre-existing data without the need for further online data acquisition. The most similar offline RL algorithm to RIBBO is Decision Transformer [2]. It represents a paradigm of upside down RL [3,4], which casts the problem of RL as conditional sequence modeling and enhances the performance of the model by incorporating return-to-go tokens. This also provides explanation for the RIBBO's ability, which employs conditional sequence modeling and incorporates regret-to-go tokens. We have revised to add more explanation. Thank you.
>
> [1] Offline Reinforcement Learning: Tutorial, Review, and Perspectives on Open Problems. arXiv 2020
>
> [2] Decision Transformer: Reinforcement Learning via Sequence Modeling. NeurIPS 2021
>
> [3] Reinforcement Learning Upside Down: Don't Predict Rewards - Just Map Them to Actions. arXiv 2019
>
> [4] RvS: What is Essential for Offline RL via Supervised Learning? ICLR 2022
>
>
> ### Q2: Justification for RTG inclusion
>
> Thanks for your question. As shown in Eq.(4), the training loss is defined as the next token prediction loss, which essentially constitutes a conditional generation task. The context data and RTG tokens constitute the conditions that guide the generation of subsequent sampled point. The inclusion of RTG enables RIBBO to automatically generate sequences of query points that are related to the user-desired regret. In fact, our comparison experiments with BC have clearly shown the benefit of including RTG. BC is an important baseline to underscore the significance of RTG tokens, because the only difference from RIBBO is that we do not feed RTG tokens for BC and train to minimize the BC loss in Eq.(2). The experimental results in Figure 2 demonstrate the superiority of RIBBO over BC. Further ablations on RTG tokens in Figure 3(c) and 3(d) have also shown the influence of RTG tokens on the model's performance.
>
>
> ### Q3: Transferability across BBO problems
>
> Good point! The transferability across BBO problems is implemented in an in-context manner. The context data, collected from the new problems, provides insights for understanding the problems at hand. These contexts are integrated to construct the inputs, thereby influencing the resulting sampled points for new problems. We have revised to add more explanation. Thank you.
>
> ### Q4: BBOB experiment clarifications
>
> The dimensionality of the BBOB problem is set to $10$, and the x-axis represents the number of evaluations conducted.
>
> ### Q5: Hyperparameter sensitivity
>
> We did ablations for some important hyperparameters of RIBBO, including the model size and the sampled subsequence length $\tau$ during training. Please see the details in Appendix F.
>
> ### Q6: Literature review addition
>
> Thanks a lot for pointing out this related work. We have revised to add some discussion about Opt-GAN in our revised version.
>
> **Thank you very much for your constructive suggestions, which have really helped improve our work. We hope that our response has addressed your concerns, but if we missed anything please let us know.**

---

### Official Review · Reviewer_azoC · 2024-11-03

**Soundness:** 3
**Presentation:** 3
**Contribution:** 3
**Rating:** 8
**Confidence:** 3

**Summary:**

Black-Box Optimization (BBO) is the process of optimizing an objective function where neither analytic expression nor derivatives of the objective function are available. Although Bayesian Optimization (BO) and Evolutionary Algorithms (EA) are the main BBO algorithms developed, they solve BBO problems from scratch while relying on expert-derived heuristics. Recently, the meta-learning paradigm has been adopted to learn some components of the algorithms from previously collected data, yet it requires a degree of expert knowledge. On the contrary, learning the entire BBO algorithm in an End-to-End (E2E) fashion from data requires no expert knowledge at all while providing flexibility among a wide range of BBO problems. However, learning such algorithms is challenging, and in practice, selecting the specific BBO algorithm is needed during inference (as in OptFormer [1]). In this work, an approach named Reinforced In-context BBO (RIBBO) is proposed to learn a general BBO algorithm from offline data collected using multiple BBO algorithms. Using a causal transformer, RIBBO can predict the next query point, given the optimization history. To alleviate the need for expert knowledge during inference when selecting the desired BBO algorithm, RIBBO augments the optimization history with regret-to-go RTG tokens representing the future performance of an algorithm. Consequently, RIBBO can automatically identify different algorithms and generate sequences of query points that satisfy the specified regret. So, it adds an interpretable component that is easy to manipulate during inference. To update the RTG tokens during testing, this work proposes a novel Hindsight Regret Relabelling (HRR) strategy by setting the immediate RTG as 0. Finally, the potential of this approach has been demonstrated in three problem domains: BBOB synthetic functions, hyperparameter optimization, and robot control problems while comparing to classical approaches (that generated the offline dataset) and related baselines like Behavior Cloning (BC) and OptFormer [1].

[1] Chen, Yutian, et al. "Towards learning universal hyperparameter optimizers with transformers." Advances in Neural Information Processing Systems 35 (2022): 32053-32068.

**Strengths:**

- The paper is well-written and clear.
- The literature is covered nicely in addition to motivating the proposed approach of learning the BBO algorithm in an E2E fashion.
- The experiments and the baselines demonstrate the potential of the proposed approach.
- The discussion section and ablation studies are comprehensive to understand the impact of each component in the algorithm.

**Weaknesses:**

There are no significant weaknesses in this work, yet some important comments should be mentioned to improve this work.
- The limitation of this approach is not well highlighted.
- The quality of plots can be improved. I personally don't like the overlapping vertical lines that represent the standard deviation. A shaded region could be better.

**Questions:**

- What are the major limitations of this approach?
- During inference, as a user, how can I select the initial RTG? Could you please provide examples where a low or a high initial RTG value is preferred? I believe I didn't get this part well.

---

> ### Author Response · Authors · 2024-11-25
> **Response by the authors**
>
> Thanks for your valuable and constructive comments. Below please find our responses.
>
> ### Q1: The major limitation of RIBBO
>
> The major limitation of this approach is the absence of a comprehensive explanation regarding the mechanisms of in-context learning and the convergence properties of the learned model. A mathematical theoretical analysis about the cumulative regret analysis and generalization analysis would be valuable for this work. Such insights will be useful for refining the algorithmic designs and enhancing overall performance. We have revised to highlight this limitation.
>
> ### Q2: Use shaded region for plots?
>
> Thanks for your suggestion. We actually tried to use shaded region in the plots, but the standard deviation for some algorithms is very large, which leads to overlapping shaded regions. The overlap results in color mixing and can alter the appearance of the original hues and render the plots visually chaotic. We will try to find more effective methods to represent the experimental results. Thanks anyway.
>
> ### Q3: How to select the initial RTG
>
> Benefiting from the proposed HRR strategy in Section 3.2, we suggest simply setting the RTG to 0 at each iteration (including the initial RTG). As the RTG represents the cumulative future regret, a value of 0 can make the algorithm able to balance exploration and exploitation automatically, which is also validated by the good empirical performance using this setting. The ablations about the influence of the RTG token are provided in Figure 3(c) and Figure 6(b), respectively.
>
> **Thank you very much for your positive review and constructive suggestions.**

---

> > ### Comment · Reviewer_azoC · 2024-12-01
> > **Acknowledgment**
> >
> > Dear Authors,
> >
> > Thank you for answering my concerns! I would like to keep my score.

---

> > > ### Author Response · Authors · 2024-12-02
> > > **Response by the authors**
> > >
> > > Thanks for your feedback! We are glad to hear that your concerns have been addressed. We will make sure to include the discussion in the final version. Thank you.

---

### Official Review · Reviewer_Zcjr · 2024-11-04

**Soundness:** 2
**Presentation:** 2
**Contribution:** 2
**Rating:** 3
**Confidence:** 3

**Summary:**

The paper proposes a method to solve black-box optimization (BBO) problems end-to-end, using a "decision transformer"-like approach.
This involves learning a transformer that given a context of query-observation pairs and regret-to-go tokens, selects the next query. This appears to learn from offline meta-training, how to build a model and use this model to make sequential decisions.

**Strengths:**

The BBO problem is a very relevant problem. Given that BBO can be framed as a special case of RL (where the state is kept constant), applying a decision transformer (DT) (or something similar) to this problem seems like an interesting approach.

**Weaknesses:**

I see some weaknesses both in the proposed method (and how it differs from DT), and the experimental evaluation.

Methodology:

* In my opinion, the manuscript would benefit from a more careful distinction from decision transformers, especially, since the BBO setting is a special case of the standard RL setting (where the state is constant). Why do the authors use regret-to-go tokens as opposed to returns-to-go in DT? In DT, no "observation tokens" are added to the sequence, and instead returns (i.e., function "observations") are subtracted from the RTG. Furthermore, I would like to see a comparison against DT as a baseline in the experiments. From the current paper, it does not become evident why RIBBO would be preferable over DT.

* The assumption that the "true" optimum value $y^*$ is known seems off to me. How would this value be known in practice, e.g., for a molecular design task? Other BBO algorithms such as UCB/EI/... do not require knowledge of $y^*$, and instead effectively balance exploration & exploitation. This seems like a fundamental assumption of the proposed approach, and to my understanding is why regret-to-go tokens might not be so practical (DT uses return-to-go tokens which do not have this problem).

* Setting $R_t = 0$ before choosing the next action suggests to me that the acquisition function is executed in the "exploitation regime". That is, in the meta-training set, I would assume that actions from algorithms that follow $R_t = 0$ are most likely from a stage where the optimum has already been almost perfectly determined, and the BBO algorithm purely exploits the model (i.e., does not focus on further exploration). Perhaps "knowing" the true $y^*$ a priori can compensate for this lack of exploration, but I would assume that this does not work in challenging exploration tasks where $y^*$ is unknown a priori. Can the authors elaborate why they expect setting $R_t = 0$ would lead to explorative behavior?

Experimental evaluation:

* It is not clear to me why "imitating" a BBO algorithm (like UCB, EI, ...) with a transformer is preferable over optimizing said algorithm directly. The "imitation" setting seems more applicable to cases where we want to imitate decisions of a "human expert" that cannot be expressed in closed-form, as is done in BC. This paper seems to claim that the main reason for learning a transformer is that this can leverage meta-training data to select the right BBO algorithm on the fly, however, in my view the experiments are insufficient to support this claim. This is because (1) the experiments do not compare against any other ad-hoc method of choosing BBO algorithms on the fly based on their performance on similar tasks in a meta-training set (this would not require re-learning the BBO algorithm, just learning when to apply which one), and (2) the experiments do not appear to compare against baselines that also use the meta-training set. In other words, I believe that the experiments are heavily skewed in RIBBO's favor since RIBBO uses a meta-training set while most BBO baselines have not seen this extra data. It might be that RIBBO exploits additional patterns in the meta-training set beyond just imitating one of the BBO algorithms. In my view, the experiments should compare against BBO algorithms that operate over meta-learned models which incorporate knowledge from the meta-training set. See, for example, [1]. In my opinion, convincing experiments would have to evaluate against such approaches that meta-learn individual components.

* It seems to me that the most important results are in Appendix D where the authors test RIBBO in scenarios where the same function family are not included in the offline meta training set. In the results from the main test, RIBBO might make extensive use of offline information that is not available to most of the tested baselines (for example, GP-EI). It seems that in Appendix D, RIBBO is frequently outperformed by GP-EI. Can the authors elaborate on why that could be? This seems concerning to me.


[1]: Rothfuss et al., Meta-Learning Reliable Priors in the Function Space.

**Questions:**

* I would suggest that before Eq. (2) it is highlighted that this is the standard behavioral cloning baseline.

* Can't the proposed HRR also lead to OOD RTGs, in case the $R_t \gg 0$ when no similar data was seen before in the meta-training set? That is, if the tasks encountered are more difficult than the tasks in the meta-train set.

* What is the size of $M$ and $T$ in the experiments? Do the results look different with smaller meta training sets? What are the minimal values where RIBBO starts to outperform basic BBO algorithms (that do not use the meta training data)?

* In Figures 3b,3c,3d, why were the specific functions chosen / "cherry-picked" for this visualization? Can you add the visualizations with the other test-functions to the supplementary material?

* In multiple places (for example, line 534), the authors claim that RIBBO can be used to generate optimization trajectories satisfying a user-specified regret. This claim does not seem to be sufficiently supported by the experiments. The experiments do indicate that there is a correlation between the user-specified regret and the obtained regret, but even in Figure 3c, the mean regret when a user specified regret $10$ is significantly larger than $10$.

---

> ### Author Response · Authors · 2024-11-25
> **Response by the authors (1/2)**
>
> Thanks for your valuable and constructive comments. Below please find our responses.
>
> ### Q1: The comparison of regret-to-go and return-to-go
>
> Good point! The benefit of using regret-to-go is its independence from a predetermined horizon length when using the HRR strategy, contrasting with the return-to-go method where the calculation is usually related to the product of the maximal achievable reward and the horizon length. Once established, the return-to-go is expected to decrease as the optimization progresses. Additionally, from the perspective of optimization, the cumulative regret provides a more meaningful measure of the performance (measuring the gap to the optimum), compared to the sum of $y$. Therefore, regret-to-go has been adopted as the preferred metric in the algorithm's design. Thanks to your suggestion, we have revised to add some discussion to make it clear. Thank you.
>
> ### Q2: A comparison against DT as a baseline in the experiments should be provided
>
> Thanks for your valuable suggestion. In fact, the comparison against DT using regret-to-go was provided in our paper. Figure 3(d) is a comparison between HRR and the naive regret updating strategy. The naive strategy in Section 3.2, is indeed the strategy used in DT, which specifies a desired performance as the initial regret $R_0$ and decreases it as $R_t = R_{t-1} - (y^*-y_t)$ using the regret at each iteration. The proposed HRR strategy consistently outperforms the naive strategy across the whole optimization stage, demonstrating the advantages of HRR over DT.
> We have revised this part to make it clearer. Thank you.
>
> ### Q3: The assumption about the ''true'' optimum value
>
> Yes, as you stated, the calculation of the regret-to-go tokens requires the ''true'' optimum value. In our experiments, we used the average of the best-observed values across the training tasks as a proxy for the optimum. Other ways may include using the maximum of the best-observed values across the training tasks.
>
> ### Q4: Setting $R_t=0$ is executed in the ''exploitation regime''
>
> Thanks for your comments. Setting $R_t=0$ will not purely exploit. As detailed in the final paragraph of Section 3.2, the immediate RTG tokens are designed to represent the performance based on cumulative regret over the future trajectories (i.e., $R_t=\sum_{t'=t+1}^T (y^* - y_{t'})$), rather than focusing solely on instantaneous regret $y^* - y_{t+1}$. Thus, setting the immediate RTG to $0$ inherently accounts for future regret, thereby enabling the algorithm to autonomously trade-off the exploration and exploitation.
>
> Sampling the next query point $x_t$ is not only influenced by the RTG token, but also the previous history $(x_i, y_i, R_i)^{t-1}\_{i=0}$. The RTG token represents the goal that is intended to be achieved, while the historical data contains information about the problem. These elements are integrated to construct the inputs, influencing the resulting sampled point. Even if the RTG token is set to 0, implying a desire to sample the optimum, the short length of the history suggests limited knowledge about the problem, prompting the model to retain explorative behavior. As the history extends, suggesting a sufficient knowledge of the problem, the preference encoded by the RTG token shifts towards more greedy action.
>
> The experiment in Table 2 has shown the variation of std of the output Gaussian head of $x$. The std is very large in the early stage and becomes small later, which demonstrates the ability to trade-off the exploration and exploitation. And we further examine the performance of RIBBO with different values of $R_t$. The results in Figure 6(b) show that setting to a value larger than $0$ will decrease the performance and converge to a worse value, validating the effectiveness of setting the immediate RTG as 0.
>
> ### Q5: Comparison to BBO algorithm selection methods
>
> Thanks for your suggestion. To our knowledge, there have been no meta-learning based BBO algorithm selection methods, that can choose appropriate BBO algorithms on the fly based on their historical performance on the meta-training set. Thus, we have revised to add the comparison with a simple yet commonly used bandit-based algorithm selection method, referred as Bandit. The probability of selecting the $i$-th algorithm in each iteration is proportional to $v_i + \sqrt{\frac{2*N}{n_i}}$, where $v_i$ is the average $y$ associated with the $i$-th algorithm, $n_i$ denotes the number of times the $i$-th algorithm has been chosen, and $N$ is the aggregate number of evaluations. The results are shown in Figure 8 in Appendix H of the revised version. The performance of Bandit is suboptimal, which may be because of the limited evaluation setting in our experiments. The evaluation allocation among different algorithms may waste evaluations on underperforming algorithms and thus result in worse overall performance.

---

> ### Author Response · Authors · 2024-11-25
> **Response by the authors (2/2)**
>
> ### Q6: Comparison to meta-learning individual components
>
> In our paper, we focus on training an end-to-end model using offline datasets and introduce RTG tokens to enhance the performance. Therefore, the most related baselines are those training an end-to-end model with offline datasets, e.g., BC and OptFormer, and thus we also only compared them.
>
> Thanks to your suggestion, we have revised to conduct experiments against FSBO [1], a method that meta-trains priors for GP and is widely used as a baseline in the meta-learning BBO studies [2,3,4]. The results are shown in Figure 8 in Appendix H of the revised version. FSBO has inferior performance to both BO and RIBBO, which may be because the methodology of meta-training priors to model the landscape may suffer from highly heterogeneous landscapes, derived through a sequence of transformations in the BBOB test functions. Even worse, an inappropriate prior might potentially degrade performance. RIBBO is designed to learn the whole optimizer, which has a greater expressive capacity and enables it to perform better in such scenarios.
>
> [1] Few-Shot Bayesian Optimization with Deep Kernel Surrogates. ICLR 2021
>
> [2] Towards Learning Universal Hyperparameter Optimizers with Transformers. NeurIPS 2022
>
> [3] Scalable Meta-Learning with Gaussian Processes. AISTATS 2024
>
> [4] Pre-trained Gaussian Processes for Bayesian Optimization. JMLR 2024
>
> ### Q7: RIBBO is frequently outperformed by GP-EI
>
> In Appendix D, the experiments focus on cross-distribution generalization, where the same function family are excluded from the offline meta training set. The incorporation of inappropriate offline information from disparate function family may affect the performance, due to the heterogeneity among these function families. Despite this, RIBBO still manages to other end-to-end methods, e.g., BC and OptFormer, demonstrating its relative advantages. But as you indicated, generalization is a very important but challenging issue in the domain of meta-learning. In the future, it is a very interesting direction to develop more robust and generalizable optimizers.
>
> ### Q8: When the new problems are more difficult, HRR leads to OOD RTGs
>
> Yes, higher RTGs may occur in OOD when problems are more difficult. However, the model trained tends to exhibit a degree of tolerance to these OOD RTGs. A similar capacity is also noted in DT, where the return-to-go tokens are typically assigned values greater than those in the training datasets. OOD issues are a common problem in meta-learning and an interesting open challenge. Thanks for pointing out this, and we have revised to mention it in the future work.
>
> ### Q9: The setting of $M$ and $T$ in the experiments
>
> Thanks for your question. $M$ is set to 100,000 for synthetic functions and $150,000$ for rover problems. For HPO-B problems, it is 25,000. $T$ is set to $150$ for synthetic and rover problems, and 100 for HPO-B problems. We have revised to make them clear.
>
> The results will be different as the size of training sets increases. In the experiments, we save the model per 100,000 steps during training and select the model with the smallest validation loss.
>
> ### Q10: Why were the specific functions chosen
>
> According to the properties of the BBOB functions, they are divided into $5$ categories, and we randomly selected one from each category for visualization. Thanks to your suggestion, we have revised to show the visualizations with the other test-functions in Figure 9 in Appendix I of the revised version.
>
> ### Q11: RIBBO can be used to generate optimization trajectories satisfying a user-specified regret？
>
> We are very sorry that we did not make it clear. As you indicated, it should be a positive correlation between the user-specified regret and the obtained regret. We have revised to make the statement more precise. Thank you very much.
>
> **Thank you very much for your constructive suggestions, which have really helped improve our work. We hope that our response has addressed your concerns, but if we missed anything please let us know.**

---

> ### Comment · Reviewer_Zcjr · 2024-11-25
>
> I thank the authors for their detailed responses to my concerns and questions, and appreciate the author's efforts during the rebuttal. I increased my score for soundness based on the latest revision.
>
> * Regarding Q2, I was referring to a comparison to DT with *return-to-go* tokens. I believe that such a comparison would be instructive, since the regret-to-go tokens are one of the main novelties of the paper. In its current form, I am not fully convinced by the arguments in favor of this approach.
> * I appreciate that the authors mention that RIBBO requires a-priori knowledge of the true optimum before sampling the function. In my view, this is a major limitation of a method tackling black-box optimization, and I feel that this should be highlighted more prominently in the manuscript.
> * I am not fully convinced by the given argument that the length of a history is a good proxy to determine if exploration was sufficient. This is clearly only the case if RIBBO is *trained on data that is very similar to the test data*. Furthermore, I still find this and the results in Appendix D, which show that RIBBO cannot robustly generalize across synthetic functions, concerning, and indicative of issues with the above points.
>
> Based on these concerns, I am inclined to keep my current score.

---

> > ### Author Response · Authors · 2024-11-28
> > **Response by the authors**
> >
> > ### Q1: A comparison against DT
> >
> > Thanks for your suggestion. We also feel very sorry that we did not understand your comment exactly. Now we have revised to add a comparison against DT with return-to-go tokens in Figure 10 in Appendix J of the revised version. For your convenience, we also put the figure in [RIBBOrebuttal](https://anonymous.4open.science/r/RIBBO/DT_comparison.png). The only difference from RIBBO is that the regret-to-go tokens of RIBBO fed into the model are substituted with return-to-go tokens. During inference, the initial return-to-go values of $50$, $100$, $150$, and $200$ are tested for DT. We can clearly observe that RIBBO achieves superior performance or at least comparable efficacy in comparison to DT. Additionally, we can observe that the performance of DT depends upon the selection of initial return-to-go values, e.g., the best performance on GriewankRosenbrock and SharpRidge occur at the initial values of $100$ and $50$, respectively. A carefully selected initial return-to-go is crucial for DT solving different functions.
> >
> > ### Q2: The limitation of RIBBO
> >
> > Thanks for your suggestion. We have revised to highlight the limitation of our method in the future work section.
> >
> > ### Q3: Generalization
> >
> > Yes, the explanation we provided may be just one potential reason for the observed performance. We think that a theoretical analysis about the mechanisms of in-context learning and the convergence properties will be helpful to provide insights into how the model balances exploration and exploitation, as well as how it utilizes the information from the collected history. However, to be honest, we cannot do it currently. We will try to do rigorous theoretical analysis in our future work.
> >
> > In our method, we only use limited information about the problem. That is, only the optimization history \{$x_{t'}, y_{t'}$\}$_{t'=1}^{t-1}$ on the problem is available to the model. To further improve the generalization, we may incorporate additional meta-features related to the problem [1,2], such as the range, type and statistical features of each variable, which can facilitate more accurate problem identification, and better exploration and exploitation trade-off. We have revised to add this in our limitation. Thank you very much.
> >
> > [1] Initializing Bayesian Hyperparameter Optimization via Meta-Learning. AAAI 2015
> >
> > [2] Towards Learning Universal Hyperparameter Optimizers with Transformers. NeurIPS 2022
> >
> > **Thank you once again for your time and valuable insights in reviewing our paper. We hope that our response can address your concerns, but if we missed anything please let us know.**

---

> > > ### Comment · Reviewer_Zcjr · 2024-12-03
> > >
> > > Thank you for your response to my concerns and for revising your submission to mention these limitations. I strongly suggest that these limitations are mentioned in the main body of the paper rather than in the appendix.
> > > Nevertheless, these limitations still stand, which makes me inclined to keep my current score.

---

### Official Review · Reviewer_REqP · 2024-11-08

**Soundness:** 4
**Presentation:** 4
**Contribution:** 3
**Rating:** 8
**Confidence:** 4

**Summary:**

This paper is about learning a black-box optimization algorithm in an end-to-end fashion. In terms of meta learning, this approach falls in the category of "meta-learning the entire algorithm". The reason for doing this is that it is more flexible than learning something that replaces heuristics in an existing black-box optimization algorithm.  The approach proposed in this paper builds on a large sequence model (causal transformer). The goal is that this model can learn from the sequential behavior of many BBO algorithms on many tasks for making decisions. The supervision signal in this learning task is the regret-to-go which captures the performance of the optimization algorithm. The training regime is offline and the training sequences consist of query points and function values. During the test phase, the model is used in an autoregressive way to generate new query points. Experiments are performed on synthetic functions / data and robotic control problems. The proposed method is compared to state-of-the-art baselines.

**Strengths:**

The paper is well-written and clear. The presentation is easy to follow and the claims and contributions are clearly stated. The method is novel and performs better than state-of-the-art approaches on the selected experiments.

**Weaknesses:**

The main weakness of this method is the fact that it has to transfer or generalize to new optimization problems, while algorithmic optimizers have to be tuned to new optimization problems. It is hard to judge how much training is needed for this method to perform well on any "similar" problem.

**Questions:**

In practice, it is also possible to just run the algorithmic optimizer longer to get better solutions and it would be interesting if this is also true for this approach.

Some algorithmic optimizers use local approximations of the objective functions to estimate higher order informations (e.g., gradients) would this be useful for this approach or does this already happen inside of the learned model?

---

> ### Author Response · Authors · 2024-11-25
> **Response by the authors**
>
> Thanks for your valuable and constructive comments. Below please find our responses.
>
> ### Q1: Generalize to new optimization problems
>
> When applied to new optimization problems, the tuning of our method is executed in an in-context manner, with the model parameters being frozen. The context data collected from the optimization of new problems provides insights for understanding the problems at hand, and influences the resulting sampled points. It is hard to exactly judge how much context data is needed. A mathematical theoretical analysis about the cumulative regret analysis and generalization analysis can be helpful and is an interesting avenue for future work to explore. Thank you for your suggestion.
>
> ### Q2: Run the algorithmic optimizer longer to get better solutions
>
> Running the algorithmic optimizer longer typically enhances the quality of the obtained solutions. In our experiments, the behavior algorithms employed a consistent horizon to collect datasets, so we simply use the same horizon for inference. In the future, it is interesting to collect optimization histories with diverse horizons to serve as training datasets to enable the algorithmic optimizer to be robust to the horizon length. Further, we can incorporate the advanced length extrapolation techniques in transformer [1] to increase the inference horizon to obtain better solutions.
>
> [1] Train Short, Test Long: Attention with Linear Biases Enables Input Length Extrapolation. ICLR 2022
>
> ### Q3: Add local approximations methods as behavior algorithms
>
> Yes, adding local approximations methods, such as gradient estimating techniques, can enhance our method. As outlined in Appendix B.2, the characteristics of the training datasets significantly affect the effectiveness of the model. It is recommended to utilize well-performing and diverse behavior algorithms for dataset generation. The integration of well-performing local approximation methods can increase the diversity of the datasets, thereby facilitating the development of a more potent learned model.
>
> **Thank you very much for your positive review and constructive suggestions. We have revised to add them as interesting future works, and will explore them in the future.**

---

### Meta-Review · Area_Chair_KcDt · 2024-12-25

**Metareview:**

This paper proposed the RIBBO method to enhance Black-Box Optimization by utilizing offline data and incorporating regret-to-go tokens, with potentials to improve the efficiency of optimization processes. The paper does not adequately address the limitations and comparisons with existing methods. The AC agrees with the reviewers that the work would benefit from a more careful distinction from decision transformer family of algorithms. It has not meet the standards expected for publication.

**Additional Comments On Reviewer Discussion:**

The reviewers opinions keeps steady during the rebuttal period. AC agrees that a substantial improvement should be made before its publication.

---

### Decision · Program_Chairs · 2025-01-22

Reject